# Weakly-Supervised Affordance Grounding Guided by Part-Level Semantic Priors

**Peiran Xu, Yadong Mu** [*]
Wangxuan Institute of Computer Technology
Peking University
Beijing, China
{xpr820,myd}@pku.edu.cn

## Abstract

In this work, we focus on the task of weakly supervised affordance grounding, where a model is trained to identify affordance regions on objects using human-object interaction images and egocentric object images without dense labels. Previous works are mostly built upon class activation maps, which are effective for semantic segmentation but may not be suitable for locating actions and functions. Leveraging recent advanced foundation models, we develop a supervised training pipeline based on pseudo labels. The pseudo labels are generated from an off-the-shelf part segmentation model, guided by a mapping from affordance to part names. Furthermore, we introduce three key enhancements to the baseline model: a label refining stage, a fine-grained feature alignment process, and a lightweight reasoning module. These techniques harness the semantic knowledge of static objects embedded in off-the-shelf foundation models to improve affordance learning, effectively bridging the gap between objects and actions. Extensive experiments demonstrate that the performance of the proposed model has achieved a breakthrough improvement over existing methods. Our codes are available at https://github.com/woyut/WSAG-PLSP .

## 1 Introduction

The concept of affordance, originating from psychology (Gibson, 1977), refers to the possibilities for action that the environment offers to an animal. In recent years, this concept has been extensively applied in the field of embodied AI (Nagarajan & Grauman, 2020; Brohan et al., 2023; Wang et al., 2024a). In particular, affordance grounding aims to locate the region on an object that affords some specific action. This task can serve as a link between visual perception and robotic control, allowing agents to focus on critical areas of objects during action execution, thereby enhancing task-specific grasping or interaction.

Early works in this field concentrate on fully supervised settings, utilizing network architectures designed for semantic segmentation. Although these methods achieve strong evaluation performance, fully supervised datasets are often limited in terms of the diversity of actions and objects (e.g., less than 10 affordance classes in Myers et al. (2015) and Nguyen et al. (2017)). This is mainly due to the high cost and inherent subjectivity or ambiguity in pixel-level affordance annotation (e.g., determining which part of *chopsticks* affords *grasping*). In response, a more practical weakly supervised setting (Luo et al., 2022) has received increasing attention. It assumes that only the image-level categories are provided as supervising signals during training, while the model needs to predict heatmaps to indicate the affordance regions during testing. Additionally, for each object and affordance category involved, the training set includes a set of "exocentric" images, which depict scenes where the object is in use. (Please see Figure 1 for some illustrations.) These images implicitly provide information about the affordance regions. Therefore, an important task for the model is to transfer the knowledge from the interaction images in the exocentric view to the static object images in the egocentric view. This problem setting effectively addresses the issue of data diversity, making it convenient to expand the scale of affordance data by grabbing images from the internet or other

---

[*]Corresponding author.

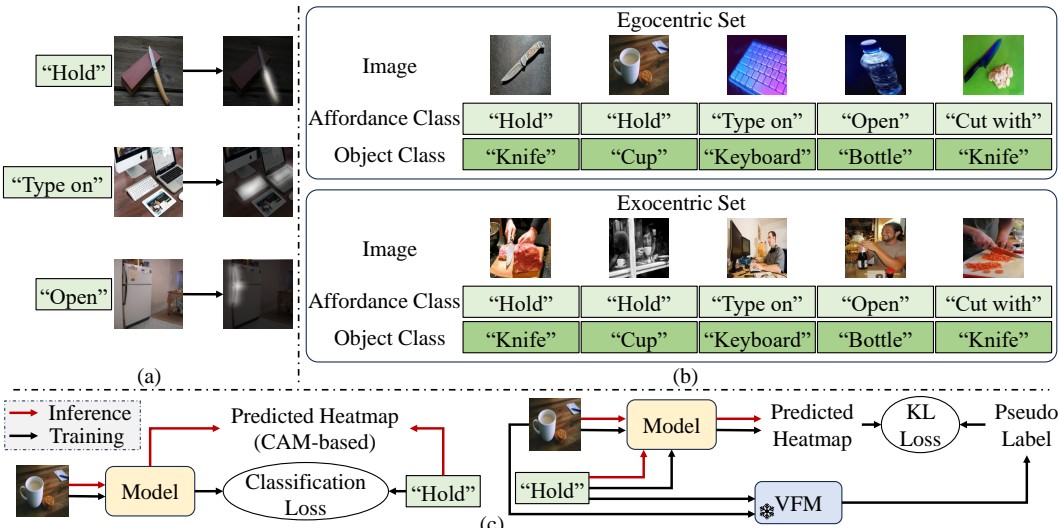

Figure 1: An overview of the WSAG task, including the visualization of the inference setting (a) and the training set (b), the comparison between the pipeline of previous CAM-based methods (c-left) and the proposed pseudo fully supervised method (c-right).

vision datasets. More importantly, learning affordance from interaction demonstrations is consistent with human cognitive processes. Through analyzing the exocentric images, the model can achieve a more natural and robust understanding of affordance regions.

To address this *weakly supervised affordance grounding* (WSAG) task, existing attempts (Luo et al., 2022; Li et al., 2023; Xu et al., 2024) are mostly based on class activation maps (CAM) (Zhou et al., 2016). They train the network to perform affordance classification (*i.e.* action classification) during training, and use the CAMs as model prediction for inference. Such a pipeline is constrained to a fixed affordance label set. Also, CAM is known to be inaccurate since it only highlights the most discriminative region in an image. In WSAG, the response area for a certain affordance may have significantly different appearances (*e.g.* open a bottle vs. open a refrigerator). So a simple classification task may have difficulty connecting these sub-image areas together, which further restricts the quality of CAMs. Essentially, for a static image of an object, affordance is more akin to a potential (the possibility of action) rather than an existence. Therefore, methods based on verb classification might have inherent limitations. On the other hand, there is also room for improvement in the utilization of exocentric images. Previous methods (Luo et al., 2022; Li et al., 2023; Zhang et al., 2024b; Chen et al., 2024a) typically incorporate a similarity loss to align the pooled exocentric and egocentric feature maps. It should be noted that the exocentric images and the egocentric images do not contain the same object instance, and the backgrounds may vary a lot between them. Thus, aligning the features obtained from global pooling (Luo et al., 2022) or masked pooling (Li et al., 2023) guided by CAM (which may be not accurate) could introduce noises unrelated to the object affordance, preventing the model from extracting the most useful knowledge for WSAG.

With the recent development of generic computer vision, visual foundation models (VFMs) and multi-modal large language models (MLLMs) are capable of providing high-quality dense annotations in a zero-shot manner, which can be utilized in downstream semantic tasks (Chen et al., 2023c; Kweon & Yoon, 2024; Chen et al., 2024b). In this work, We try to unleash the power of these advanced models in the field of affordance. Specifically, we create a mapping from action-related affordance classes to object/part queries, and use these queries to generate pseudo labels for the egocentric images. The pseudo labels enable a supervised learning pipeline to approach the WSAG task, which shows great performance improvement compared to CAM-based methods. However, the naive pseudo labels still possess some drawbacks and may fail to locate the affordance region in some hard cases. To avoid overfitting to the imperfect labels and further optimize the performance, we introduce several extra modules to the baseline model. In particular, the exocentric images in the training set are leveraged to regularize the training process through cross-view feature alignment.

Unlike previous works, by turning to the VFMs again, we precisely confine the alignment within the region of the target object being interacted with, with a focus on affordance-related information. On the other hand, we note that the occlusion of objects by humans in the exocentric images can serve as a reliable cue for affordance regions. Based on this observation, an additional label refinement stage is designed to provide better pseudo labels than the initial ones. Besides, to equip the model with a stronger ability to handle unseen object categories during inference, we incorporate a lightweight reasoning module to help the model better understand different manifestations of the same affordance across different objects.

To sum up, our contributions are as follows. (1) We propose a pipeline to generate affordance heatmaps using off-the-shelf visual foundation models, and introduce the first pseudo-supervised training framework for the task of WSAG. (2) We devise multiple techniques to enhance the performance of the baseline model, including pseudo label refinement, object feature alignment with exocentric images, and a reasoning module to upgrade the generalization ability. (3) We conduct extensive experiments to validate the effectiveness of the proposed method, whose performance achieves a breakthrough improvement compared with previous works.

## 2 RELATED WORKS

### 2.1 AFFORDANCE LEARNING

As an important concept in embodied intelligence, object affordance learning has been studied in various contexts. To be specific, research in this field starts from 2D affordance grounding. Several fully supervised datasets (Myers et al., 2015; Roy & Todorovic, 2016; Nguyen et al., 2017; Luddecke & Worgotter, 2017; Shah & Khalifa, 2023; Guo et al., 2023) have been built for image or RGBD input, and models for object detection or semantic segmentation have been applied (Nguyen et al., 2016; Do et al., 2018; Caselles-Dupré et al., 2021; Mur-Labadia et al., 2023). Some works have extended this task by relaxing the supervision to key points (Sawatzky et al., 2017), handling few-shot learning scenarios (Luo et al., 2021; Hadjivelichkov et al., 2023; Li et al., 2024a), employing multi-task training (Chen et al., 2022; Qian & Fouhey, 2023), and leveraging human poses (Luo et al., 2023b). On the other hand, Deng et al. (2021),Yang et al. (2023),Nguyen et al. (2023),Delitzas et al. (2024) and Yang et al. (2024) consider 3D affordance grounding on point clouds. Some works (Mo et al., 2021; Wang et al., 2022; Wu et al., 2022; Liang et al., 2023; Li et al., 2024b) further incorporate this into a manipulation system to perform active learning. Besides, there are also lines of works on affordance-based action anticipation (Fang et al., 2018; Nagarajan et al., 2019; Liu et al., 2022; Chen et al., 2023a; Bahl et al., 2023; Yu et al., 2023) and hand/pose generation (Corona et al., 2020; Ye et al., 2023; Wang et al., 2024b; Kim et al., 2024).

This work mainly focuses on a weakly supervised setting of 2D affordance grounding proposed in Luo et al. (2022), as it has the potential to efficiently scale the size and diversity of data and mimics how humans perceive affordances. Previous works are mostly built upon CAMs (Zhou et al., 2016). Luo et al. (2022) and Luo et al. (2023a) design an invariance mining module to extract useful knowledge in the exocentric images, while aligning the globally pooled ego-/exo-centric image features. Li et al. (2023) aims to focus on the target object parts. It performs localized alignment between the masked pooled egocentric features and the clustering center of the exocentric features. Xu et al. (2024) adds the affordance label to model input and introduces several attention modules to perform inter- and intra-modal interactions. Zhang et al. (2024b) defines a modified version of WSAG that simultaneously predicts the action category, object category, and affordance heatmap. All these methods lack direct supervision on the predicted heatmap, while we design a pipeline harnessing the pseudo labels generated by foundation models. Some existing works have also realized the value of foundation models. Li et al. (2024a) fintunes DINOv2 (Oquab et al., 2024) on a one-shot dataset. Cuttano et al. (2024) finetunes CLIP (Radford et al., 2021) on a referring object segmentation task and directly applies it to affordance grounding. Huang et al. (2024) finetunes an LLM to generate the bounding box of the affordance region. Qian et al. (2024) finetunes LLaVA (Liu et al., 2024) to perform affordance reasoning and introduces an additional decoder to generate heatmap. Tong et al. (2024) utilizes an LLM followed by a VLM to produce the affordance segment. Ju et al. (2024) leverages pretrained CLIP and a diffusion model to perform contact point detection based on an external memory. These works either rely on strong supervision or adopt zero-shot inference, both of which are in stark contrast to our approach of weakly supervised training utilizing image-level

labels and exocentric images. More similarly, Chen et al. (2024a) and Rai et al. (2024) incorporate frozen CLIP and SAM (Kirillov et al., 2023) into the training pipeline, but they still follow Li et al. (2023)'s pipeline, thus share its limitations.

## 2.2 WEAKLY SUPERVISED SEMANTIC SEGMENTATION

As a closely related topic, weakly supervised semantic segmentation (WSSS) aims to perform segmentation with only image-level category labels. A significant body of research has been conducted based on CAM, while leveraging additional guidance from foundation models has gained increasing popularity. For example, Xie et al. (2022), Lin et al. (2023), Xu et al. (2023), and Zhang et al. (2024a) make use of the rich semantics in CLIP to generate or refine the CAMs. Recently, the generic segmentation model SAM and its variants (like Ren et al. (2024)) benefit WSSS more straightforwardly. They can be used to post-process the generated CAMs (Chen et al., 2024c; Yang & Gong, 2024), produce direct supervision for CAMs when training the classifier (Kweon & Yoon, 2024; Yang & Gong, 2024), or completely replace CAMs to generate pseudo labels (Sun et al., 2023c; Jiang & Yang, 2023; Chen & Sun, 2023). The feature of our work is extending the idea of foundation-model-based pseudo labeling to the field of affordance, using models designed for recognizing objects to help with a task related to actions and interactions.

## 3 METHOD

### 3.1 TASK FORMULATION

We follow the WSAG setting proposed in Luo et al. (2022), as shown in Figure 1. During training, we have access to a set of egocentric images $I^{\text{ego}}$ (*i.e.* the images of static objects) and exocentric images $I^{\text{exo}}$ (*i.e.* the images of objects in interaction). For each ego-/exo-centric image $I$, we have its semantic label $o$ (*i.e.* object class) and affordance label $a$ (*i.e.* action class). At the inference stage, we are required to generate an affordance heatmap $H$ given an egocentric image and an affordance label. We formulate this task in an open-vocabulary fashion, which means $a$ can be free-form text and is not constrained to a fixed set during inference.

### 3.2 MODEL ARCHITECTURE

From a high-level perspective, our model can be divided into four modules: (1) a visual encoder $\text{Enc}_V$ that transforms an image $I$ into a feature map $F_V \in \mathbb{R}^{d \times h \times w}$; (2) a textual encoder $\text{Enc}_T$ that transforms the affordance query $a$ into a feature vector $f_T \in \mathbb{R}^d$; (3) a cross-modal fuser that interacts $f_T$ with $F_V$ and generates a feature vector $f_A \in \mathbb{R}^d$ representing the grounding target; (4) a decoder Dec that outputs the predicted heatmap $H_{\text{pred}}$ based on $F_V$ and $f_A$.

We adopt CLIP (Radford et al., 2021) (ViT-B/16) as the visual and textual encoder due to its rich semantic information, which can provide useful prior knowledge for affordance learning. The decoder basically follows the design of SAM's (Kirillov et al., 2023) mask decoder that uses two-way Transformer blocks and deconvolutions to generate the prediction, and it will be introduced in detail in appendix A. The cross-model fuser is implemented as several Transformer blocks that perform cross-attention using the text feature as the query and the image feature as the key and value. Overall, we keep a plain model and focus on the design of supervision instead. More advanced architectures can be easily introduced into the model and may further enhance its performance.

### 3.3 GENERATING PSEUDO LABELS

The basic idea of our method is to leverage visual foundation models to generate pseudo labels $H_{\text{pl}}$ for the egocentric images, and then train a grounding model in a supervised manner. Two steps are involved in this labeling process: determining a grounding target and performing the grounding task. To begin with, we observe that existing visual models typically excel at understanding semantic concepts, such as object categories and attributes (noun or adjective). In contrast, an affordance query often refers to an action or function (verb), making it challenging to directly use $a$ as the grounding target to obtain reliable pseudo labels. For a given object $o$, the region corresponding to a specific type of affordance is usually one or several semantic parts of the object.

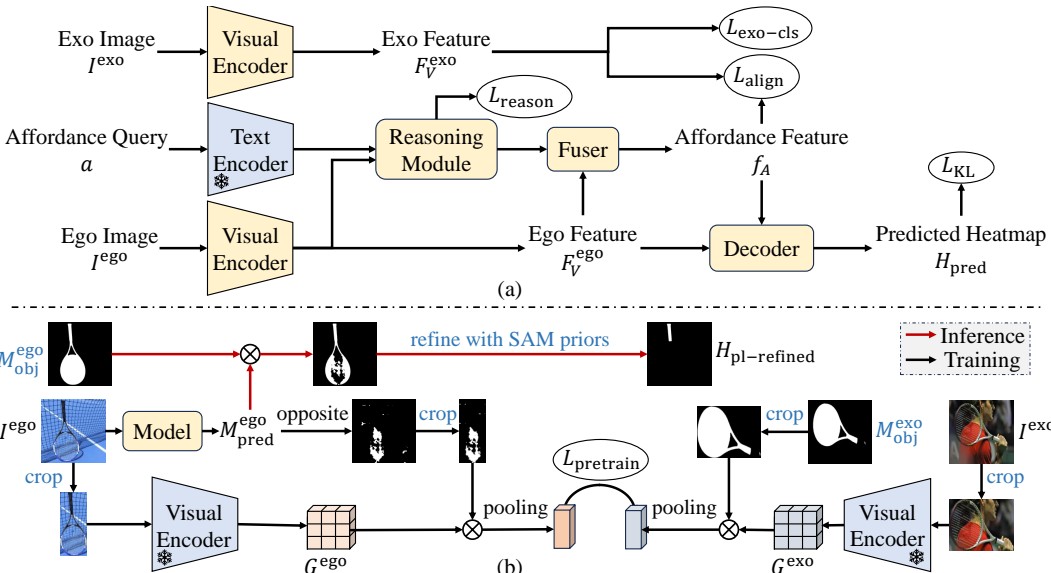

Figure 2: (a) The training pipeline of the proposed method. (b) The label refinement stage described in Section 3.5. The inputs/operations marked in blue font are generated/performed using the foundation models (VLpart and SAM).

Therefore, we first define a mapping $P(o, a) = p$ that converts the combination of an object category and an affordance category into a part of the object. The part name, $p$, can be easily understood by an off-the-shelf foundational model, thereby serving as a proper grounding target. For example, $P(\text{"knife"}, \text{"hold"}) = \text{"handle of the knife"}$, $P(\text{"bottle"}, \text{"open"}) = \text{"cap of the bottle"}$. The part name mapping $P$ can be implemented using an LLM that integrates extensive common-sense knowledge (Tong et al. (2024)), while we manually define it in this work, since there are only around 100 possible combinations of $o$ and $a$ in current datasets. The details of the $P$ we build are shared in Appendix B. Note that $P$ is only used for generating pseudo labels, and is not called during inference.

For the second step, we need a model for part-level, open-vocabulary semantic grounding. We find the project of Sun et al. (2023a) suitable for this task. It first detects the bounding box of the object part referred by the text input using VLpart (Sun et al., 2023b). Then, SAM (Kirillov et al., 2023) is employed to generate a segmentation mask using the box as the prompt. We devise some heuristic rules to prevent SAM from treating the background as the foreground (details are provided in Appendix B). Having obtained the segmentation mask $M_{\text{part}}^{\text{ego}}$, we transform it into a heatmap that acts as the pseudo label $H_{\text{pl}}$. We use the KL-divergence $L_{\text{KL}}$ between $H_{\text{pred}}$ and $H_{\text{pl}}$ as loss function to train the model. As will be shown in Section 4.3, this baseline method has already achieved good performance compared with previous weakly-supervised methods.

However, the pseudo labels generated using this simple pipeline are far from perfect. Specifically, (1) VLpart is not familiar with certain part names, and sometimes gives a mask of the whole object instead of the target part. (2) Some affordance regions cannot be recognized as a well-defined part, thus we cannot find an accurate text input for VLpart. These issues constrain the quality of the pseudo labels, and limit the precision of model predictions as a result of supervised training. In the following subsections, we will introduce several methods that we devise to improve the baseline model.

## 3.4 ALIGNING WITH EXOCENTRIC FEATURES

The training process described above totally ignores the exocentric images in the training set, so the first improvement we propose is to utilize exocentric images to regularize the feature learning process. Given an exocentric image that has the same object class and affordance class as the egocentric image, we first pass it into the visual encoder $\text{Enc}_V$. Let us denote the feature maps

of the egocentric image and the exocentric image as $F_V^{\text{ego}}$ and $F_V^{\text{exo}}$, respectively. To leverage the implicit guidance provided in the exocentric images, previous methods like Luo et al. (2022) directly align $F_V^{\text{ego}}$ and $F_V^{\text{exo}}$ after global average pooling (GAP). Ideally, the alignment should draw close the features of the static object in the egocentric image to the features of the object in use in the exocentric image. However, the feature vectors obtained through GAP are mixed with irrelevant information, such as the background in egocentric images and the human in exocentric images. These interferences may lead the alignment process into confusion. Li et al. (2023), Chen et al. (2024a), and Zhang et al. (2024b) further employ heatmaps predicted by the model to restrict the feature alignment within a specific region. However, due to the lack of direct supervision on the predicted masks under the weakly supervised setting, it cannot be ensured that they actually focus on the target object.

To alleviate this problem, we once again utilize VLpart and SAM to locate the image region for alignment. Following the procedure in Section 3.3, we generate an object mask $M_{\text{obj}}^{\text{exo}}$ for each exocentric image using the object class $o$ as query. Based on this, we perform masked average pooling (MAP) to get $f_E \in \mathbb{R}^d$, a feature of the active object, and align the grounding target in our pipeline to it. Formally,

$$f_E = \text{Avg-Pool}(M_{\text{obj}}^{\text{exo}} \cdot F_V^{\text{exo}}), \tag{1}$$

$$L_{\text{align}} = 1 - \text{Cos-Sim}(f_A, \text{stop-grad}(f_E)), \tag{2}$$

where the stop gradient operation is used to facilitate better supervision on the egocentric branch during the alignment process. Meanwhile, we incorporate a lightweight classification head $\text{Head}_{\text{exo}}$ to perform affordance classification on $f_E$:

$$L_{\text{exo-cls}} = \text{CE}(\text{Head}_{\text{exo}}(f_E), \hat{a}), \tag{3}$$

where CE is the cross entropy loss, and $\hat{a}$ is the one-hot version of $a$. Through $L_{\text{exo-cls}}$, we encourage $f_E$ to carry information about human-object interactions. Consequently, aligning $f_E$ with $f_A$ will help $f_A$ focus on the part of the object that is related to the interaction, *i.e.*, the affordance region.

Previous methods often use multiple exocentric images to accompany one egocentric image, but we do not see a clear advantage of using more than one exocentric image for the alignment in our pipeline. So we only pair each egocentric image with one exocentric image to reduce the computational burden. Additionally, since different instances of the same object category can exhibit significantly different appearances, we believe that it is beneficial to filter the set of exocentric images used for feature alignment. For each egocentric image $I^{\text{ego}}$, instead of randomly sampling one exocentric image of the same object and affordance class, we first identify a candidate pool of exocentric images containing object instances that are most similar to the instance in $I^{\text{ego}}$. Here the similarity score $s$ is defined as the cosine similarity of the egocentric image's feature map and the exocentric image's feature map after MAP, *i.e.*

$$s = \text{Cos-Sim}(\text{Avg-Pool}(M_{\text{obj}}^{\text{exo}} \cdot F_V^{\text{exo}}), \text{Avg-Pool}(M_{\text{obj}}^{\text{ego}} \cdot F_V^{\text{ego}})), \tag{4}$$

where $M_{\text{obj}}^{\text{ego}}$ is the object mask of the $I^{\text{ego}}$ obtained using the same method as $M_{\text{obj}}^{\text{exo}}$. The pool size $N_{\text{exo-pool}}$ is set to 10 in our experiment. This filtering process further eliminates irrelevant information, enabling the feature alignment to better concentrate on the transfer of affordance knowledge. Note that the calculation of similarity and the formation of the candidate set can be done offline, so it will not bring additional computation to the training process.

## 3.5 REFINING PSEUDO LABELS

In addition to guiding feature alignment, exocentric images can also be directly used to refine the sub-optimal pseudo labels generated by VLpart and SAM. As shown in Figure 1, in an image of human-object interaction, the interaction region of the object (which corresponds to the affordance region in the egocentric image) is usually occluded by the human body. Based on this clue, we propose a novel pseudo label refinement stage prior to the affordance learning stage described above. We utilize the same model architecture as Section 3.2, except that the last softmax layer is replaced by sigmoid to produce a mask $M_{\text{pred}}^{\text{ego}}$ ranged in $[0, 1]$ for $I^{\text{ego}}$. Next, a separate, frozen feature extractor $\text{Enc}_V'$ (*e.g.* a pretrained CLIP or DINO encoder) is employed to get the feature map $G^{\text{ego/exo}}$ of the object crop in the egocentric image and the exocentric image:

$$G^{\text{ego/exo}} = \text{Enc}_V'(\text{Crop}(I^{\text{ego/exo}}, b^{\text{ego/exo}})), \tag{5}$$

where Crop means to crop an image using a bounding box, and $b^{\text{ego/exo}}$ is the bounding box generated by VLpart using $o$ as query. Finally, we introduce a loss function based on semantic similarity to guide the training process:

$$L_{\text{pretrain}} = 1 - \text{Cos-Sim}(\text{Avg-Pool}(\tilde{M}^{\text{ego}} \cdot G^{\text{ego}}), \text{Avg-Pool}(\tilde{M}^{\text{exo}} \cdot G^{\text{exo}})), \quad (6)$$

where $\tilde{M}^{\text{ego}} = \text{Crop}(1 - M^{\text{ego}}_{\text{pred}}, b^{\text{ego}})$, $\tilde{M}^{\text{exo}} = \text{Crop}(M^{\text{exo}}_{\text{obj}}, b^{\text{exo}})$. If the predicted mask $M^{\text{ego}}_{\text{pred}}$ manages to distinguish the affordance region, then $\tilde{M}^{\text{ego}}$ will activate the remaining part of the object, and the masked object feature $\tilde{M}^{\text{ego}} \cdot G^{\text{ego}}$ will be similar to the occluded object feature $\tilde{M}^{\text{exo}} \cdot G^{\text{exo}}$, resulting in low loss value. The pipeline of the refinement stage is shown in Figure 2(b).

After this preparatory stage, the mask $M^{\text{ego}}_{\text{pred}} \cdot M^{\text{ego}}_{\text{obj}}$ can be used as a better pseudo label for affordance grounding when $H_{\text{pl}}$ is flawed. To further reduce the noise and make the shape and boundary of the pseudo label more explainable, a post-processing session is performed based on SAM. Specifically, we employ SAM's Auto Mask Generator (*i.e.* using grid points as prompt) to parse $I^{\text{ego}}$ into small segments and find the segments that have a large intersection area with $M^{\text{ego}}_{\text{pred}} \cdot M^{\text{ego}}_{\text{obj}}$. The union of these selected segments forms the final pseudo label $H_{\text{pl-refined}}$ for the training of the next stage. Some examples of the label refinement stage's output are shown in Figure 3.

A straightforward idea is to incorporate the loss $L_{\text{pretrain}}$ into the affordance grounding training stage like $L_{\text{align}}$. However, we empirically find it hard to balance $L_{\text{pretrain}}$ and the $L_{\text{KL}}$. The output heatmap is sometimes highly concentrated at a few points (thus $\tilde{M}^{\text{ego}}$ nearly covers the whole object), which may result in low overall loss but is not desired. Thus, we use $L_{\text{pretrain}}$ to lead a separate training phase, and further refine its output using the segmentation priors of SAM. More details related to this preparatory phase can be found in Appendix B.

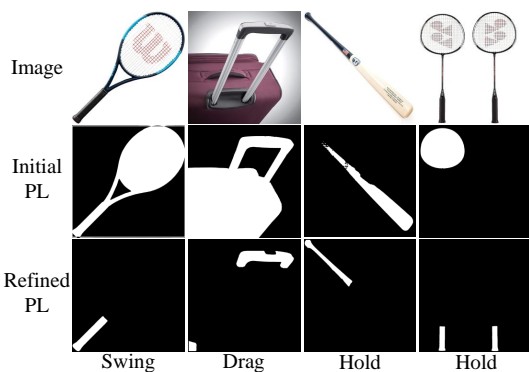

Figure 3: A visualization of the effects of the label refinement phase.

### 3.6 HANDLING UNSEEN OBJECTS

Another problem of the baseline method lies in its generalization ability towards unseen object categories. Since the size and diversity of the existing training dataset are still quite limited, it is hard to generalize a certain affordance to novel object classes during testing. For instance, the model may fail to identify how to hold a cup if it only encounters holding bottles while training. To tackle this issue, we propose to introduce a simple yet effective reasoning module that explicitly helps the model build the relationship between object $o$ and affordance $a$. In more detail, we first take $I^{\text{ego}}$'s class token $c_V \in \mathbb{R}^d$ generated by $\text{Enc}_V$, and send it to a shallow MLP to predict the object class. Then, another MLP is responsible for predicting the target part given the predicted object class and the affordance query. Cosine similarity between the predicted object/part and the annotated object/part category is employed as the objective to train these two MLPs. Formally,

$$\begin{aligned}
f_{\text{pred-obj}} &= \text{MLP}_{\text{noun}}(c_V), \\
f_{\text{pred-part}} &= \text{MLP}_{\text{part}}([f_{\text{pred-obj}}, f_T]), \\
L_{\text{reason}} &= (1 - \text{Cos-Sim}(f_{\text{pred-part}}, \text{Enc}_T(o))) + 0.1 \cdot (1 - \text{Cos-Sim}(f_{\text{pred-obj}}, \text{Enc}_T(p))),
\end{aligned} \quad (7)$$

where $[f_{\text{pred-obj}}, f_T]$ is the concatenation of $f_{\text{pred-obj}}$ and $f_T$ along the feature dimension. The similarity is calculated in the feature space of $\text{Enc}_T$ (*i.e.* CLIP text encoder, see Section 3.2). The coefficient 0.1 is used to balance part learning and object learning. Embedded with information of the target part, $f_{\text{pred-part}}$ is added to $f_T$, and the sum is sent to the cross-modal fuser and the decoder for heatmap generation.

Conceptually, this reasoning module can be viewed as learning to approximate the mapping $P$ (as in Section 3.3), transforming relatively abstract actions into part names that the model can comprehend

Table 1: The results on AGD20K. The best and second-best results are marked as **bold** and underline, respectively.

| | Seen Split | | | Unseen Split | | |
|---|---|---|---|---|---|---|
| | KLD↓ | SIM↑ | NSS↑ | KLD↓ | SIM↑ | NSS↑ |
| Cross-View-AG (Luo et al., 2022) | 1.538 | 0.334 | 0.927 | 1.787 | 0.285 | 0.829 |
| Cross-View-AG+ (Luo et al., 2023a) | 1.489 | 0.342 | 0.981 | 1.765 | 0.279 | 0.882 |
| LOCATE (Li et al., 2023) | 1.226 | 0.401 | 1.177 | 1.405 | 0.372 | 1.157 |
| WSMA (Xu et al., 2024) | 1.176 | 0.416 | 1.247 | 1.335 | 0.382 | 1.220 |
| Strategies (Rai et al., 2024) | 1.194 | 0.400 | 1.223 | 1.407 | 0.362 | 1.170 |
| WorldAfford (Chen et al., 2024a) | 1.201 | 0.406 | 1.255 | 1.393 | 0.380 | 1.225 |
| Ours-baseline (Sec 3.3) | 0.938 | 0.503 | 1.477 | 1.256 | 0.428 | 1.346 |
| Ours-full | **0.890** | **0.510** | **1.547** | **1.153** | **0.437** | **1.418** |

more readily. Notably, here the parts are represented in the form of latent features rather than text, facilitating the processing of affordance regions that are difficult to articulate in natural language.

Besides, we notice that the model sometimes does not fully utilize the textual input $a$. This could be attributed to the fact that most objects in the dataset are annotated with only one type of affordance, thus the model can give good predictions in many cases relying solely on the information from the visual branch (especially on the unseen split of AGD20K dataset, which will be detailed in the next section). In such scenarios, $L_{reason}$ may be meaningless since the model almost ignores $f_A$ when generating affordance heatmaps. To make the model leverage the text input more effectively, we design a simple "stitching" augmentation during training. For each egocentric image $I^{ego}$, we randomly sample 3 irrelevant egocentric images and stitch the 4 images into a new one in a $2 \times 2$ layout, where $I^{ego}$ may appear at any of the 4 positions. Given this combined image as visual input, the model has to rely on the text query to distinguish the target object and target region. This augmentation is randomly applied to the input image with a probability of 0.5.

Combining all techniques mentioned above, our full model is trained with

$$L_{all} = L_{KL} + \lambda_1 (L_{align} + L_{exo-cls}) + \lambda_2 L_{reason}, \tag{8}$$

where the supervision for $L_{KL}$ is the refined label $H_{pl-refined}$. The full pipeline is shown in Figure 2(a).

## 4 EXPERIMENT

### 4.1 DATASET AND METRICS

We conduct experiments on AGD20K (Luo et al., 2022), a widely-used WSAG dataset. It contains about 30K exocentric and egocentric images, annotated with 36 affordance classes and 50 object classes. Following prior arts, two types of data split are considered in the experiments. For the seen split, all object categories exist in the training set. While for the unseen split, there is no intersection between the object categories in the training set and the test set, meaning that the model needs to generalize the knowledge of affordance to novel objects during testing. The evaluation metrics include Kullback-Leibler Divergence (KLD), Similarity (SIM), and Normalized Scanpath Saliency (NSS), which are consistent with previous works. Following the implementation of Luo et al. (2022) and Li et al. (2023), the evaluation is conducted at the resolution of $224 \times 224$.

### 4.2 IMPLEMENTATION DETAILS

We train the model for 40 epochs using the AdamW optimizer. The learning rate is set to 1e-4 and the batch size is set to 20. The learning rate of the CLIP visual encoder is reduced to 1e-5 to prevent losing important semantic information acquired during its pre-training stage, while the CLIP text encoder is frozen. The loss coefficients $\lambda_1$ and $\lambda_2$ are set to 10 and 1, respectively. We use random cropping, random flip, and the stitching technique mentioned in Section 3.6 as augmentation. In order to reduce the impact of randomness, the reported performance is the averaged result across five independent training sessions using different random seeds.

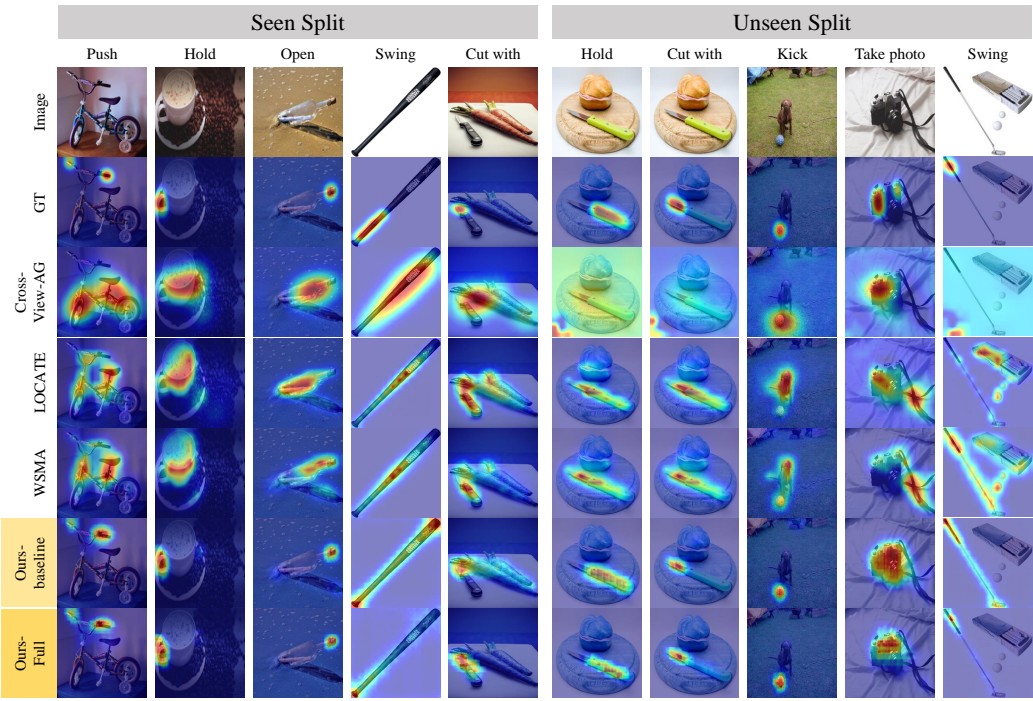

Figure 4: Qualitative results of our model and previous methods.

Table 2: The analysis of the proposed modules. The best and second-best results are marked as **bold** and underline, respectively.

| refinement Sec 3.5 | alignment Sec 3.4 | reasoning Sec 3.6 | Seen Split | | | Unseen Split | | |
|---|---|---|---|---|---|---|---|---|
| | | | KLD↓ | SIM↑ | NSS↑ | KLD↓ | SIM↑ | NSS↑ |
| | | | 0.938 | 0.503 | 1.477 | 1.256 | 0.428 | 1.346 |
| ✓ | | | 0.916 | 0.506 | 1.524 | 1.246 | 0.427 | 1.336 |
| | ✓ | | 0.924 | 0.507 | 1.486 | 1.176 | **0.437** | 1.407 |
| | | ✓ | 0.943 | 0.508 | 1.464 | 1.242 | 0.433 | 1.353 |
| ✓ | ✓ | ✓ | **0.890** | **0.510** | **1.547** | **1.153** | **0.437** | **1.418** |

## 4.3 MAIN RESULTS

As shown in Table 1, our baseline method, where $H_{\text{pl}}$ serves as supervision and $L_{\text{KL}}$ is solely used for training, has already shown significant improvement over previous methods. The results demonstrate the benefit of shifting from a weakly supervised setting to a pseudo-fully supervised setting. Integrating the refined labels, the feature alignment loss, and the reasoning module into the pipeline, our full model further achieves consistently better results than the baseline model on all metrics. Figure 4 visualizes some predicted affordance heatmaps. It can be seen that our method produces heatmaps that are more precise and more focused. Particularly, for an image of knife in the unseen split, previous methods produce similar heatmaps for "hold" and "cut with", while our model clearly distinguishes the different affordance queries. For difficult cases like "swing (baseball bat)"/"cut with (knife)" in the seen split and "swing 9golf club)" in the unseen split, the baseline model may fail due to sub-optimal pseudo labels or interfering objects, while the full model gives better prediction thanks to the improved pipeline.

## 4.4 ANALYSIS ON MODULES

We perform ablation studies to examine the effectiveness of each proposed module, and the results are shown in Table 2. For the seen split, both the label refinement stage and the cross-view feature

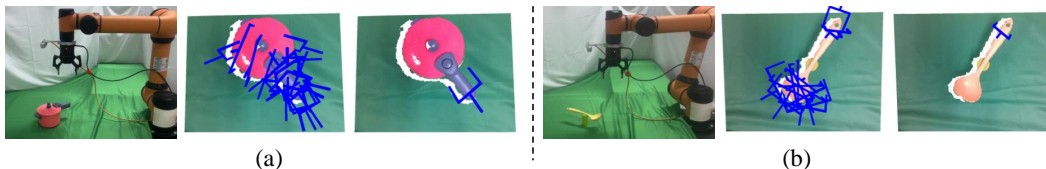

Figure 5: Two examples for the real-world experiments: (a) grasping a pressure cooker, (b) grasping a spoon. We show the reconstructed point cloud of the scene and all the poses generated by an off-the-shelf grasping algorithm, as well as the selected pose based on affordance.

alignment loss bring clear improvements to the baseline model, showing that we successfully produce better pseudo labels in the refinement stage and the exocentric images contribute to affordance feature learning. The reasoning module does not show much advantage, which is understandable since the model has encountered all kinds of objects during training and the test set does not demand cross-category generalization. While for the unseen split, where generalization ability is essential, including the reasoning loss effectively enhances the performance. Also, aligning with exocentric images leads to better outcomes by ensuring a more robust affordance learning process. The utility of the refinement stage is not that evident for the unseen split. This is because the modifications on the training labels may not influence the performance on the test set directly, as the training and test sets consist of objects that belong to totally different categories. In summary, each of the three modules has yielded the intuitively anticipated effect. Finally, combining them together further boosts the overall performance for both splits. More experimental results can be found in Appendix D.

## 4.5 DEPLOYMENT

To examine the generalization ability of our method in real-world tasks, we combine the trained affordance grounding model with a grasp planning algorithm and deploy them on a robotic arm. Specifically, we employ an off-the-shelf robotic grasping model (Ma et al., 2024) to generate grasping pose candidates. Meanwhile, we send the RGB observation into our pre-trained affordance grounding model and generate the affordance heatmap. We then choose the pose whose target point has the highest probability in the heatmap, *i.e.* utilizing affordance information to determine the most suitable grasping pose.

In Figure 5, we show some illustrative examples for the deployment experiment. We run the affordance-guided grasping pose generation pipeline to grasp a toy pressure cooker and a spoon. Both objects do not exist in the training set, and the affordance query word, "grasp", is not seen by the model during training either. In such a scenario that requires cross-category generalization, our affordance grounding model manages to select the appropriate pose and leads to successful graspings. More examples and details of the deployment will be provided in Appendix D.4.

## 5 CONCLUSION

In this paper, we address the problem of affordance grounding. We transform the traditional CAM-based weakly supervised training approach into a fully supervised training process leveraging pseudo labels. Visual foundation models are utilized to generate pseudo labels, refine them, and facilitate fine-grained feature alignment. Also, we propose a simple yet effective data augmentation method and a noun/part reasoning module to improve the model's generalization capability. In summary, we take a step towards applying the semantic knowledge about static objects to the learning of affordances or actions. As for future works, we believe that expanding the scale and diversity of the affordance dataset is a direction worth exploring, since the pseudo labeling technique greatly reduces the burden of data annotation and exhibits satisfactory performance. We would also attempt to upgrade the reasoning module to enhance the generalization on unseen objects, especially those that have vastly different structures with training categories. This could be achieved through leveraging the common sense knowledge in LLMs or knowledge bases.

ACKNOWLEDGMENTS

The work was supported by National Key R&D Program of China (2022ZD0160300), an internal grant of Peking University (2024JK28), and a grant from Kuaishou (No. DJHL-20240809-115).

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

# A DETAILS OF THE MODEL

## A.1 ARCHITECTURE

As stated in Section 3.2, the proposed model consists of four modules. **The visual encoder** is a plain Vision Transformer following the ViT-B/16 in Dosovitskiy et al. (2021). It has 12 layers. The patch size, hidden dimension, and number of heads are set to 16, 768, and 12, respectively. We use the pre-trained weights provided by CLIP (Radford et al., 2021). CLIP adds a layer normalization and a linear projection to the class token which is then used for contrastive learning. Since we need to perform dense prediction, we apply these two layers to all the patch tokens, resulting in a $512d$ feature map.

**The text encoder** is also a plain Transformer. It has 12 layers. The hidden dimension and number of heads are set to 512 and 8. We use CLIP's pretrained weights and it is frozen during the whole training process. In fact, since there are only tens of different text segments to be encoded in the experiments (including verbs, class names, and part names), we pre-extract their CLIP features and discard the text encoder during training.

**The cross-modal fuser** follows the design of the regressor block in CAE (Chen et al., 2024d). Each block takes two sets of tokens as input, using one set as the query and the other as the key and value. The query tokens are updated through a cross-attention-based Transformer layer. Our fuser module comprises four such blocks, where the text token $f_T$ serves as the initial query, and the visual encoder's output (including the class token $c_V$ and the patch tokens) serves as the key and value. When the reasoning module is incorporated, $c_V$ and $f_T$ are first processed by several linear layers to produce the predicted part name feature $f_{\text{pred-part}}$ (Eq (6)). Then, $f_{\text{pred-part}}$ is added to $f_T$ and the sum serves as the initial query of the fuser.

**The mask decoder** is adapted from the decoder of SAM (Kirillov et al., 2023). It first employs a two-way Transformer to perform interactions between the affordance feature $f_A$ and the feature map of the egocentric image $F_V^{\text{ego}}$. Then, it upsamples the resulting feature map and predicts the affordance mask using a dynamic classifier. In more detail, a two-way transformer block takes two sets of tokens as input, denoted as $A$ and $B$. The block consists of four layers, namely: (1) self-attention of $A$, (2) cross-attention of $A$ to $B$, (3) MLP on $A$, and (4) cross-attention of $B$ to $A$. We stack two such blocks in the decoder. The patch tokens in $F_V^{\text{ego}}$ are used as $B$, while the affordance feature $f_A$, the egocentric class token $c_V$, and a newly introduced learnable token $x$ are concatenated together to form $A$. After the transformer, we use several convolutional and deconvolutional layers to gradually upsample the output feature map (*i.e.* the output related to $F_V^{\text{ego}}$) from $14 \times 14$ (*i.e.* $\frac{224}{16} \times \frac{224}{16}$) to $56 \times 56$. We also apply an MLP to the output token related to $x$, and use it as the weight of the final dynamic classifier.

## A.2 LOSS FUNCTIONS AND TRAINING SCHEME

Four loss functions are involved in the training of the full model. The **KL loss** $L_{\text{KL}}$ aims to directly supervise the predicted heatmap. The output of the mask decoder is a logits map of shape $56 \times 56$. We first bilinearly upsample it to $224 \times 224$, then use the softmax function to transform it into a 2D heatmap, which is suitable for calculating the KL divergence. On the other hand, the generated pseudo label is a binary mask. We bilinearly interpolate it to $224 \times 224$, binarize it using a threshold of $0.5$, apply Gaussian blur to it, and finally normalize it to a 2D heatmap by dividing its sum.

The **exocentric classification loss** $L_{\text{exo-cls}}$ is simply a cross entropy loss applied to the pooled exocentric feature map $f_E$ (Eq (1)). The object mask $M_{\text{obj}}^{\text{exo}}$ should have the same shape with the feature map $F_V^{\text{exo}}$. We define it as a $14 \times 14$ binary map in which a cell equals 1 if its corresponding $16 \times 16$ patch has an intersection with the object's bounding box (detected by VLpart).

The **cross-view alignment loss** $L_{\text{align}}$ and **reasoning loss** $L_{\text{reason}}$ are both based on cosine similarity. For the former, we set a margin of $0.1$ that makes the loss 0 when the similarity is higher than 0.9.

To balance the losses above, we set $\lambda_1$ (the coefficient of $L_{\text{align}}$ and $L_{\text{exo-cls}}$) to 10 and $\lambda_2$ (the coefficient of $L_{\text{reason}}$) to 1. The code is implemented in PyTorch. We train the model for 40 epochs using the AdamW optimizer, with the learning rate set to 1e-4, betas set to (0.9, 0.95), and weight decay coefficient set to 0.01. The learning rate of the visual encoder is reduced to 1e-5 to prevent

losing important semantic information acquired during CLIP's pre-training stage. The batch size is 20. Each of the 20 egocentric images is accompanied by an exocentric image. We apply random cropping (from $256 \times 256$ to $224 \times 224$) and random horizontal flip (with a probability of 50%) to each egocentric image. The stitching augmentation mentioned in Section 3.6 is also applied. In order to reduce the impact of randomness, the reported performance is the averaged result across five independent training sessions using different random seeds (1/10/100/1000/10000).

## A.3 EVALUATION METRICS

The evaluation is performed at the scale of $224 \times 224$ in consistency with prior works (Luo et al., 2022; Li et al., 2023). The evaluation metrics are KL Divergence (KLD), Similarity (SIM), and Normalized Scanpath Saliency (NSS) (Peters et al., 2005). We follow the implementation of Luo et al. (2022). KLD is the same as the KL loss in the training stage. SIM calculates the histogram intersection of two heatmaps. For NSS, we first linearly normalize the predicted heatmap to make it have zero mean and unit standard deviation. Then we linearly normalize the groundtruth heatmap to $[0, 1]$, and binarize it using a threshold of 0.1. The resulting binary mask serves as a "fixation map", and the metric is the average value of the (normalized) predicted map at the fixation locations (where the fixation map has a value of 1).

## A.4 MODEL EFFICIENCY

We compare the statistics of our model and some recent works, and the results are shown in Table 3. The number of parameters in our model is at the same level as Cross-View-AG (Luo et al., 2022) and WSMA (Xu et al., 2024), while its computational cost is lower than them. Table 4 further illustrates the distribution of parameters and computations across the modules of our model (some lightweight MLPs in the reasoning module and the exocentric classifier are ignored). It can be seen that the visual encoder occupies a substantial portion of the parameters, and the calculation is also concentrated in the encoding stages. LOCATE (Li et al., 2023) employs a frozen DINO-ViT-S as its visual encoder, which is a smaller version of ViT compared to our CLIP-ViT-B. This explains the gap between it and our model in terms of model size and computational cost.[1]

In Table 3, we also compare the inference time across different models. It is evaluated on an NVIDIA GeForce RTX 2080Ti, and the batch size is set to 1. Thanks to the concise model structure, our method brings performance gains without significantly increasing the inference cost.

Note that all the previous works mentioned here do not support open-vocabulary affordance query, thus we do not include the text encoder in our model when comparing statistics and efficiency, which is roughly the same size as the visual encoder.

Table 3: Model statistics.

| Model | #params | FLOPs | inference time | fps |
|---|---|---|---|---|
| Cross-View-AG (Luo et al., 2022) | 120.0M | 29.1G | 0.019 | $\sim 52$ |
| LOCATE (Li et al., 2023) | 28.2M | 5.1G | 0.012 | $\sim 85$ |
| WSMA (Xu et al., 2024) | 90.2M | 79.6G | 0.033 | $\sim 30$ |
| Ours | 112.5M | 18.9G | 0.020 | $\sim 49$ |

Table 4: Detailed statistics of the proposed model.

| Module | #params | FLOPs |
|---|---|---|
| Visual Encoder | 86.2M | 17.7G |
| Cross-Modal Fuser | 12.7M | 0.4G |
| Mask Decoder | 12.2M | 0.8G |
| Total | 112.5M | 18.9G |

---

[1]We tried to replace the encoder of LOCATE with DINO-ViT-B and re-train the model, but did not see clear performance improvement.

Table 5: Some examples of designing the part name mapping with an LLM (GPT-4o). The prompt we used is: *Assume you are a professional data annotator. I will provide pairs of verbs and nouns, where the verb represents an affordance, and the noun represents an object category. I would like you to identify the part of the object that is associated with the given affordance. For example, for "hold knife", you should output "the handle of the knife"; for "open bottle", you should output "the cap of the bottle".*

| Verb | Object | Part |
|---|---|---|
| beat | drum | the drumhead of the drum |
| pick_up | suitcase | the handle of the suitcase |
| push | bicycle | the handlebars of the bicycle |
| swing | tennis_racket | the handle of the tennis racket |
| hold | wine_glass | the stem of the wine glass |

As for training, it takes about 1h to generate the initial pseudo labels for the training set. The label refinement stage requires approximately 1h, followed by around 3h for supervised training. The whole training process can be performed on a single NVIDIA GeForce RTX 2080Ti. As reference, LOCATE's training scheme takes about 7h in the same environment, while WSMA takes about 2h. Thus, the training cost of our method is basically at the same level with previous methods. Besides, our supervised training process is more straightforward than previous works, involving neither the clustering operations used by LOCATE nor the non-negative matrix factorization employed in WSMA.

## B    DETAILS OF THE PSEUDO LABEL

### B.1    POSSIBLE WAYS TO GENERATE PSEUDO LABELS

Based on the advanced foundation models, we consider the following methods for generating box-level pseudo-labels: (1) directly using a multimodal LLM (*e.g.* MiniGPT-v2 (Chen et al., 2023b)); (2) directly using a foundation model specialized for detection (*e.g.* VLpart (Sun et al., 2023b)); (3) combining a detection model with label mapping. We show examples of these three pipelines in Figure 6 and 7. From the pilot experiments, we observed that the output of method (1) is occasionally unstable and might be sensitive to prompts. Method (2) also demonstrates poor results since the grounding model is not familiar with affordance. The third method effectively bridges the gap between affordance and object while also being efficient and convenient, thus we employ it to detect the target bounding box. As mentioned in Section 3.3, the mapping from (affordance, object) to part name can be created by hand or with the help of an LLM, and some examples are shown in Table 5.

Given the detected box of the target object part, we employ SAM (Kirillov et al., 2023) to obtain the pixel-level pseudo label. We basically follow the practice of the "Grounded Segment Part" project (Sun et al., 2023a) but make a few adaptations. To be specific, we first filter the detected boxes with a confidence score threshold of 0.5. If no box meets this standard, then we keep the box with the maximum confidence score and discard the others. The remained boxes are treated as box prompts and sent to SAM along with the egocentric image. SAM will output a binary mask for the target object part. However, since the box prompt is not expressive enough, the output mask may, at times, confuse the foreground with the background. To handle this problem, we devise a simple sanity check that calculates the sum of the mask's value at the edges of each prompt box. Empirically, the target object (part) always lies in the center of the box and the boundary is mostly fulfilled by background. Thus, if the sum exceeds some threshold (which we set to half of the perimeter of the box), then the mask is likely to highlight the background instead of the foreground. In such circumstances, we invert the mask value inside the box ($0 \rightarrow 1$ and $1 \rightarrow 0$).

In contrast to the two-stage annotation pipeline (box and mask), LISA (Lai et al., 2024) designs an end-to-end MLLM specialized for reasoning segmentation. It takes an image with a text query and directly outputs a mask for the target region. In Figure 8, we also show some of its predictions for affordance queries. It does not exhibit an advantage over the VLpart+SAM pipeline in our task.

As mentioned in Section 3.3, our annotation process based on VLpart and SAM still gives sub-optimal labels sometimes. Some mistakes only occur accidentally on the difficult samples (*e.g.*

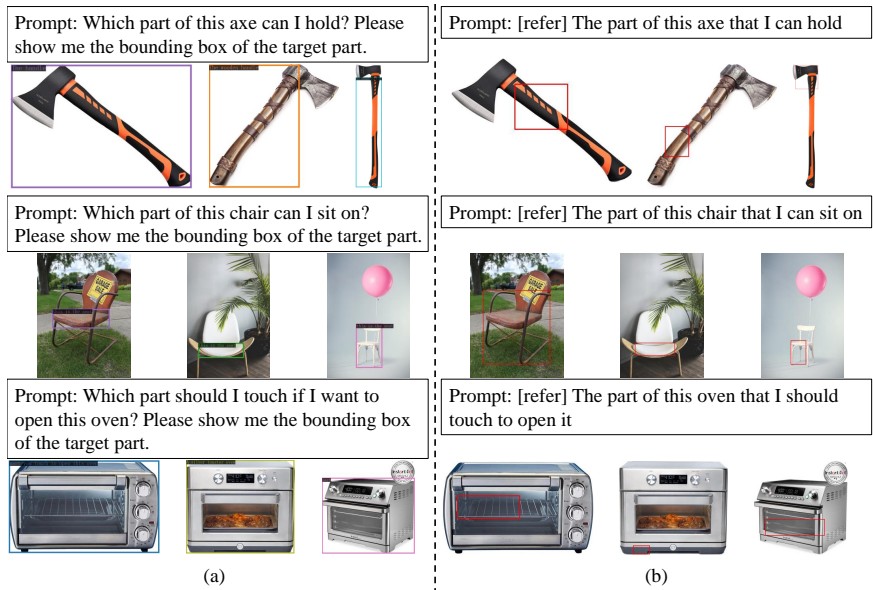

Figure 6: Examples for different pseudo label generation methods. (a) The result of Qwen-VL (Bai et al., 2023), a generic MLLM that supports image understanding. We use the model deployed at QianWen web portal (https://tongyi.aliyun.com/qianwen, version 2.5). (b) The results of MiniGPT-v2 (Chen et al., 2023b), an MLLM tailored for vision and language tasks like visual question answering and object grounding. We use the MiniGPTv2(7B)-chat model weights (after stage-3) provided at its official repo (https://github.com/Vision-CAIR/MiniGPT-4). Though both models excel at detecting objects and parts by their names or attributes, they fail to fully comprehend the affordance-related query in many cases. Also, some of the results are unstable across multiple runs, and may vary a lot when the query is slightly changed.

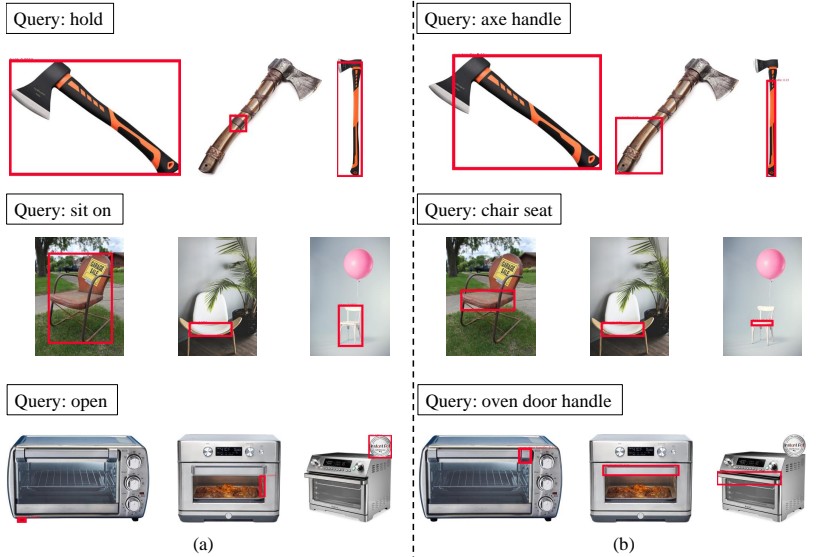

Figure 7: Examples for different pseudo label generation methods. (a) The result of an open-vocabulary (part) detection model, VLPart (Sun et al., 2023b), with affordance query. (b) The results of VLPart with part query. The segmentation model cannot fully comprehend the affordance category and usually gives oversized (object-level instead of part-level) boxes, while the part name query can usually lead to better results. The overall label quality of (b) is the best among the three types of methods mentioned in Section B.1, though it still encounters a few imperfect cases like the axe in the middle and the oven on the left.

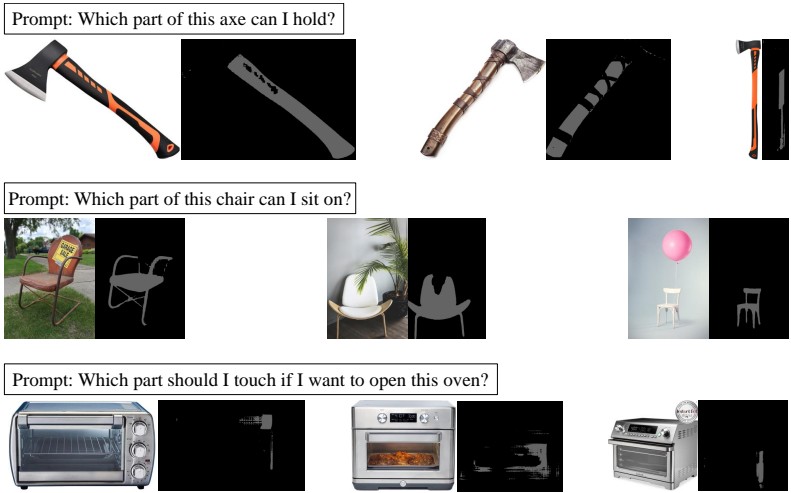

Figure 8: Some results of LISA (Lai et al., 2024). It directly outputs the segmentation mask after reasoning about affordance, but the overall performance does not show a clear advantage over our label generation pipeline.

VLpart only detects one instance when there are multiple instances in the image, SAM highlights some background regions). Other issues are more severe and exist in almost all the images across some category (*e.g.* VLpart struggles to detect the handle of a tennis racket). The former will not harm the training process greatly since the model is able to learn correct behaviors from other similar samples. While the latter clearly leads to wrong predictions and motivates us to introduce an additional label refinement stage. Appendix D.2 will show more examples about the pseudo labels.

### B.2 DETAILS OF THE REFINEMENT STAGE

The model we used for the pseudo label refinement stage is basically the same as the model for affordance grounding (described in Appendix A). The differences are as follows. (1) The exocentric branch is discarded. (2) The text input is the part name $p$ instead of the affordance class $a$, in order to offer a more straightforward and concrete signal for mask generation. Consequently, the multi-modal fuser and the reasoning module are no longer needed. The original affordance feature $f_A$ sent to the mask decoder is replaced by $c_V + \text{Enc}_T(p)$, *i.e.* the sum of the egocentric image's class token and the CLIP feature of the part name. (3) The output logits map is processed with a sigmoid function instead of a softmax function, generating a mask ranged in $[0, 1]$. In addition, the refinement stage is conducted on the affordance and object classes where our baseline model has low performance.

The mask-guided semantic similarity loss $L_{\text{pretrain}}$ is solely used as training objective. We use a frozen CLIP (ViT/B-16) as the separate visual encoder $\text{Enc}'_V$. The cropped image is padded to square and resized to $224 \times 224$ before sending to this encoder (Eq (5)). The cropped exocentric object mask $\tilde{M}^{\text{exo}}$ and predicted mask $\tilde{M}^{\text{ego}}$ are both padded to square and resized to $14 \times 14$ (*i.e.* the shape of $\text{Enc}'_V$'s output). $\tilde{M}^{\text{exo}}$ is then binarized with a threshold of 0.5. We accompany each egocentric image with 3 exocentric images, in contrast to using only 1 exocentric image in the affordance grounding stage (Section 3.4). This is because the label refinement stage lacks direct, pixel-level supervision on the predicted masks, and using more exocentric images may make the training process more robust. We compute $L_{\text{pretrain}}$ between the egocentric image and each of the three exocentric images, and use the average of them as the final loss. We utilize the AdamW optimizer with the same configuration as in Appendix A.2 to train the model for 20 epochs. Random cropping and horizontal flipping are used as augmentation.

After training the model, $M^{\text{ego}}_{\text{pred}} \cdot M^{\text{ego}}_{\text{obj}}$ can be used as a better pseudo label for affordance grounding. To further refine the mask, we call the Automatic Mask Generator of SAM, with $8 \times 8$ gird points as prompt. It will segment the whole image into non-overlapping regions, and the minimal region area

is set to 100. We then compute the intersection ratio $\frac{\text{intersection area with } M^{\text{ego}}_{\text{pred}} \cdot M^{\text{ego}}_{\text{obj}}}{\text{region area}}$ for each mask region, and select all the regions whose intersection ratio is larger than $\max(0.1, 0.9 \cdot \text{max\_ratio})$ (max\_ratio is the maximum intersection ratio among all regions). The final refined label is the union of all the selected mask regions. If there is no region selected, then we still use the original pseudo label as in Appendix B.1. Here, SAM functions as a super-pixel parser that amends the generated labels with boundary priors.

## C  COMPARISON WITH ADDITIONAL BASELINES

We give a more detailed review of the previous works related to affordance grounding, especially the AGD20K dataset (Luo et al., 2022). These works can be classified into 4 settings.

**Weakly supervised training**. This is the standard WSAG setting of AGD20K, and has been introduced in Section 3.1. Cross-View-AG (Luo et al., 2022) collected the AGD20K dataset. It designs a matrix-factorization-based invariance mining module to process the feature maps of the exocentric images, and applies global pooling to obtain an exocentric feature vector. Similarly, the egocentric feature vector is the globally pooled egocentric feature map. Both features are used for affordance classification and are aligned with each other. During inference, the CAMs of the egocentric classifier are used as model predictions. Upon this method, Cross-View-AG+ (Luo et al., 2023a) further decomposes the egocentric feature map with the same feature dictionary as the exocentric branch to improve feature transfer. LOCATE (Li et al., 2023) aims to achieve localized feature alignment. Based on a frozen DINO (Caron et al., 2021) encoder, it performs clustering to the exocentric patch tokens and selects the cluster prototype that is most likely to represent the target object based on the predicted saliency map. This prototype feature serves as the alignment target for the egocentric feature, which is the CAM-guided masked pooling result of the egocentric feature map. WSMA (Xu et al., 2024) includes the text feature of the affordance query into the pipeline. It performs contrastive learning and cross-modal fusion between the text branch and the egocentric branch. It also aligns the predicted CAM of the egocentric and the exocentric branches during training. SEA (Zhang et al., 2024b) proposes a slightly different task setting where the model simultaneously predicts the affordance heatmap, affordance category, and object category. It first fuses the exocentric features and the egocentric features to determine the category, then utilizes it to generate heatmaps. WorldAfford (Chen et al., 2024a) is built upon LOCATE. However, its input image is masked to preserve only the target object, and this object mask is generated by a zero-shot segmentation pipeline with CLIP and SAM. It also modified the structure of the DINO encoder by inserting a novel weighted context broadcasting layer. Rai et al. (2024) also employs CLIP to enhance the performance of LOCATE. It first uses an LLM to generate a description of the target affordance region, then sends this sentence to CLIP. The CLIP text feature is used to generate activation heatmaps for the egocentric and exocentric images. The exocentric heatmaps are utilized for localized clustering of the exocentric patch tokens, while the egocentric heatmap is used as supervision for the predicted CAM. In summary, all these methods give predictions based on CAM, in contrast to our pseudo supervised pipeline. Also, we propose novel alignment and reasoning modules to enhance affordance feature learning.

**Zero-shot inference**. It means directly applying a model trained on other sources of data to the affordance grounding task on AGD20K. For example, AffordanceCLIP (Cuttano et al., 2024) trains a lightweight Feature Pyramid Network (FPN) with frozen CLIP on referring image segmentation. Testing the model's performance on AGD20K, it suggests that CLIP inherently contains some interaction-related knowledge. Another method, OVAL-Prompt (Tong et al., 2024) is totally training-free. It leverages an LLM to reason about the affordance query and translate it into a part name. Then it uses an open-vocabulary segmentation model to get the mask of the target part, and send it to a grasping engine for affordance-guided manipulation. OVAL-Prompt's pipeline is very similar to our pseudo label generation pipeline. However, our contributions lie in proposing a set of strategies to refine the pseudo-labels and designing a training process based on cross-view alignment and part reasoning. Our method effectively leverages weakly supervised images, resulting in a more stable model compared to zero-shot methods. ManipVQA (Huang et al., 2024) is another LLM-based method. It unifies several robotics-related tasks into a VQA format, and constructs an instruction dataset to tune an MLLM (AGD20K is not included). Given an affordance query, it will output the coordinates of a bounding box, which is sent to a SAM-like model to generate an affordance mask. More recently, RoboABC (Ju et al., 2024) focuses on the "hold" affordance. It first builds a memory bank based on egocentric videos, which contains images of diverse objects

Table 6: The results of previous works on AGD20K under different settings. Hotspots and AffCorrs are adapted baseline methods introduced in Cross-View-AG and LOCATE, respectively.

| | Seen Split | | | Unseen Split | | |
|---|---|---|---|---|---|---|
| | KLD↓ | SIM↑ | NSS↑ | KLD↓ | SIM↑ | NSS↑ |
| *Weakly supervised training* | | | | | | |
| Hotspots | | | | | | |
| (Nagarajan et al., 2019; Luo et al., 2022) | 1.773 | 0.278 | 0.615 | 1.994 | 0.237 | 0.577 |
| Cross-View-AG (Luo et al., 2022) | 1.538 | 0.334 | 0.927 | 1.787 | 0.285 | 0.829 |
| Cross-View-AG+ (Luo et al., 2023a) | 1.489 | 0.342 | 0.981 | 1.765 | 0.279 | 0.882 |
| AffCorrs | | | | | | |
| (Hadjivelichkov et al., 2023; Li et al., 2023) | 1.407 | 0.359 | 1.026 | 1.618 | 0.348 | 1.021 |
| LOCATE (Li et al., 2023) | 1.226 | 0.401 | 1.177 | 1.405 | 0.372 | 1.157 |
| WSMA (Xu et al., 2024) | 1.176 | 0.416 | 1.247 | 1.335 | 0.382 | 1.220 |
| Strategies (Rai et al., 2024) | 1.194 | 0.400 | 1.223 | 1.407 | 0.362 | 1.170 |
| WorldAfford (Chen et al., 2024a) | 1.201 | 0.406 | 1.255 | 1.393 | 0.380 | 1.225 |
| Ours-baseline (Sec 3.3) | 0.938 | 0.503 | 1.477 | 1.256 | 0.428 | 1.346 |
| Ours-full | 0.890 | 0.510 | 1.547 | 1.153 | 0.437 | 1.418 |
| *Zero-shot inference* | | | | | | |
| AffordanceCLIP (Cuttano et al., 2024) | 1.628 | 0.335 | 0.791 | 1.812 | 0.301 | 0.760 |
| OVAL-Prompt (Tong et al., 2024) | 10.649 | 0.339 | 1.044 | 8.832 | 0.365 | 0.925 |
| *Supervised training* | | | | | | |
| AffordanceLLM (Qian et al., 2024) | - | - | - | 1.463 | 0.377 | 1.070 |
| *Few-shot supervised training* | | | | | | |
| OOAL (Li et al., 2024a) | 0.740 | 0.577 | 1.745 | 1.070 | 0.461 | 1.503 |

and their corresponding hand-object interaction points. Then, for an input image, it searches for a similar image in the memory and utilizes some semantic correspondence method to locate the key point in the input image, which can be used for downstream grasping programs. It is evaluated using point-based metrics on a subset of AGD20K with some extra object classes.

**Supervised training**. AffordanceLLM (Qian et al., 2024) considers fully supervised training on AGD20K. It divides the original test set of AGD20K (which is labeled with groundtruth heatmap) into a supervised training set (1135 egocentric images) and a smaller test set (540 egocentric images). As for the model, it sends the image feature and the affordance query into an LLM. The LLM is finetuned to predict a special token, and a mask decoder generates the affordance heatmap based on the image feature and this token.

**Few-shot supervised training**. Recently, Li et al. (2024a) proposed a One-shot Open Affordance Learning (OOAL) task. It allows access to a few supervised data (one image for each object category with manually labeled heatmaps) during training. It devises a model upon the CLIP text encoder, the DINOv2 (Oquab et al., 2024) image encoder, and a masked cross-attention transformer decoder. Specifically, the model incorporates learnable prompts in the text encoder, and multi-layer feature fusion in the image encoder. It is also observed in its experiments that DINOv2, compared with other image encoders like CLIP, embeds more part-relevant information in its representations.

Table 6 shows the experimental results of all the methods mentioned above. Our method shows a clear advantage over most methods. Notably, it does not rely on any ground-truth heatmaps, and has a significantly smaller size (∼100M parameters) than the LLM-based methods. However, our method still lags behind OOAL, though under different training settings. We note that: (1) the contributions of our work and Li et al. (2024a) are orthogonal. We focus on annotating the unlabeled egocentric images, leveraging exocentric images for alignment, and reasoning between objects and affordances, while keeping a concise model architecture. The modeling techniques used in OOAL, such as prompt tuning and DINO-based multi-layer feature fusion, can be introduced into our method and may further improve its performance. (2) As a one-shot learning method, OOAL's performance might be sensitive to the choice of the training sample. Our model may be more robust with a relatively large amount of egocentric and exocentric images that can be easily obtained.

Table 7: The results on AGD20K's "hard" split (Qian et al., 2024) The best and second-best results are marked as **bold** and underline, respectively.

| | Supervision | Hard Split | | |
| --- | --- | --- | --- | --- |
| | | KLD↓ | SIM↑ | NSS↑ |
| Cross-View-AG (Luo et al., 2022) | Weak | 2.092 | 0.209 | 0.138 |
| Cross-View-AG+ (Luo et al., 2023a) | Weak | 2.034 | 0.218 | 0.342 |
| LOCATE (Li et al., 2023) | Weak | 1.829 | 0.282 | 0.276 |
| LOCATE-Sup (Li et al., 2023; Qian et al., 2024) | Full | 2.003 | 0.224 | 0.435 |
| LOCATE-Sup-OWL | | | | |
| (Li et al., 2023; Qian et al., 2024; Minderer et al., 2022) | Full | 2.127 | 0.206 | 0.314 |
| 3DOI (Qian & Fouhey, 2023) | Zero-shot | 4.017 | 0.200 | 0.549 |
| AffordanceLLM (Qian et al., 2024) | Full | 1.661 | **0.361** | 0.947 |
| ManipVQA (Huang et al., 2024) | Zero-shot | 12.67 | 0.246 | **1.735** |
| Ours-baseline (Sec 3.3) | Weak | 1.402 | 0.353 | 1.122 |
| Ours-full | Weak | **1.395** | 0.354 | 1.151 |

# D  MORE EXPERIMENTAL RESULTS

## D.1  EXPERIMENTS ON THE "HARD" SPLIT OF AGD20K

AffordanceLLM (Qian et al., 2024) has defined a new split for AGD20K, namely the "hard" split, which is similar to the unseen split but requires a higher degree of generalization. Following the fully supervised setting (Appendix C), it has 868/807 images with dense annotations for training/testing. There is no overlap between the object categories in the training set and the test set, and the training categories share less similarity with the test categories (compared with the unseen split). Also, there are novel affordance categories in the test set, meaning that the model has to generalize among verbs as well as nouns.

We examine the performance of our model on the hard split and the results are shown in Table 7. Our full model outperforms AffordanceLLM on the KLD and NSS metrics but is slightly inferior to it on the SIM metric. On the other hand, ManipVQA performs exceptionally well on the NSS metric. The good performance of AffordanceLLM and ManipVQA on the hard split is likely attributed to the powerful reasoning capability from the LLM. We plan to explore this idea in future research.

## D.2  MORE VISUALIZATION RESULTS

In Figure 9 and 10, we give more visualization results for our generated pseudo labels, focusing on the limitations of the initial labels in particular. Besides, Figure 11 shows more qualitative results of model predictions, extending Figure 4 in the main text. We also show some typical failure cases in Figure 12.

## D.3  MORE ABLATION EXPERIMENTS

In this subsection, we conduct fine-grained ablation experiments on the proposed modules and design choices, as an extension of Section 4.4. First, we validate the effect of using the exocentric object masks $M_{\text{obj}}^{\text{exo}}$ in the alignment process. Table 8 shows that the cross-view feature alignment is not helpful without the masks, and focusing on the object area has significant advantages over globally pooling the exocentric feature maps, which is also mentioned by previous works like Li et al. (2023).

Secondly, we examine the necessity of SAM-based post-processing in the refinement stage. The results in Table 9 indicate that using the post-processed labels is essential for enhancing the model performance, which is also reflected in Figure 10.

Thirdly, we analyze the design of the reasoning module. As shown in Table 10, introducing the two reasoning losses can improve the performance on the KLD metric, but has negative effects on the

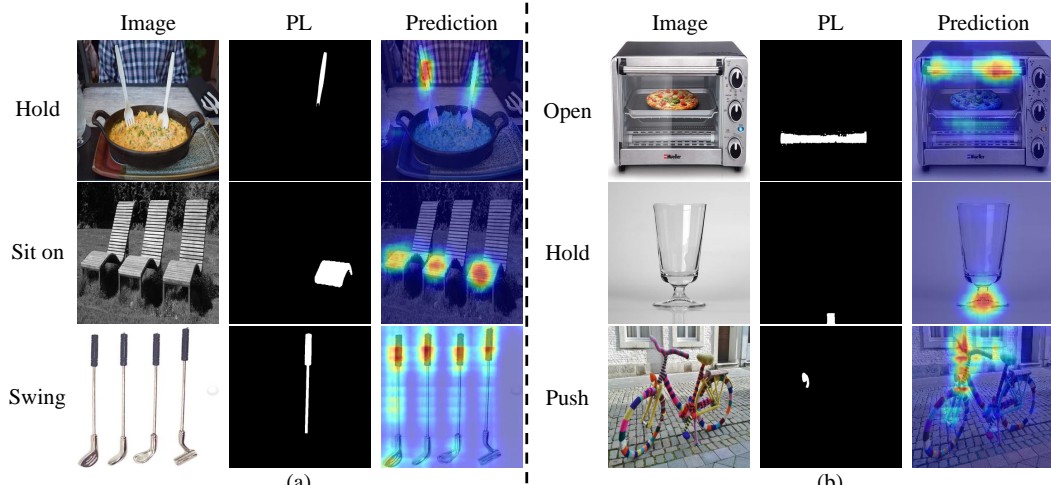

Figure 9: Visualization of the initial pseudo labels and our base model's predictions. All samples come from the training set of the seen split. (a) Though some part instances are missing in the pseudo label (due to the low confidence of VLPart's predictions), the trained model can recognize them. (b) Though the pseudo labels deviate away from the correct region in some hard cases, the trained model can output roughly satisfactory predictions. To sum up, the learned model can fix some occasional errors of the pseudo labels, when more samples in the same category have relatively high-quality pseudo labels. This is also an important advantage of our model over the zero-shot methods.

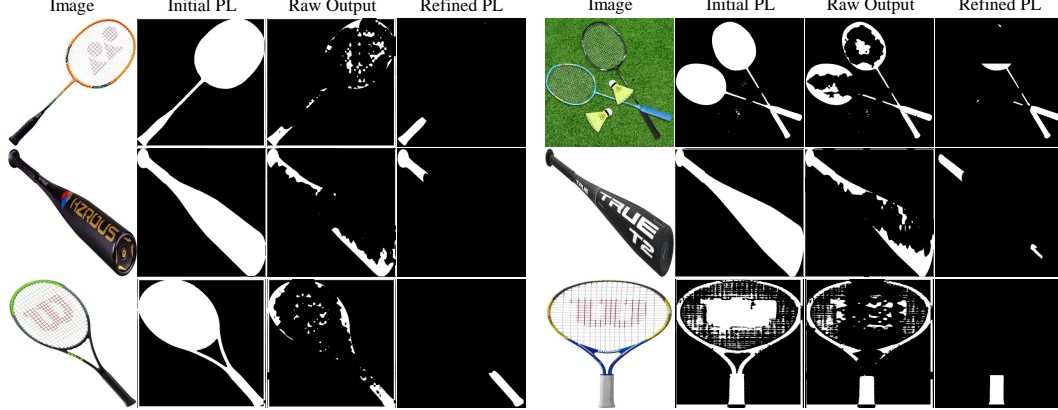

Figure 10: Visualization of the label refinement process. Here we focus on the difficult object classes for VLPart, where its prediction is inaccurate (oversized) on almost all the samples, in contrast to the examples in Figure 9. We present the initial label, the predicted result of the model in the refinement stage ($M_{pred}^{ego} \cdot M_{obj}^{ego}$), and the final label refined with SAM priors. It can be observed that the refinement stage better constrains the label to the target part, and the SAM post-processing greatly reduces the noise in the raw outputs. All these samples use "hold" as the affordance query.

Table 8: Ablation study on the alignment module. The results are obtained on the unseen split where the alignment module has a more significant effect.

| | KLD↓ | SIM↑ | NSS↑ |
|---|---|---|---|
| Ours-baseline | 1.256 | 0.428 | 1.346 |
| +alignment w.o. obj mask | 1.259 | 0.422 | 1.312 |
| +alignment w. obj mask | **1.176** | **0.437** | **1.407** |

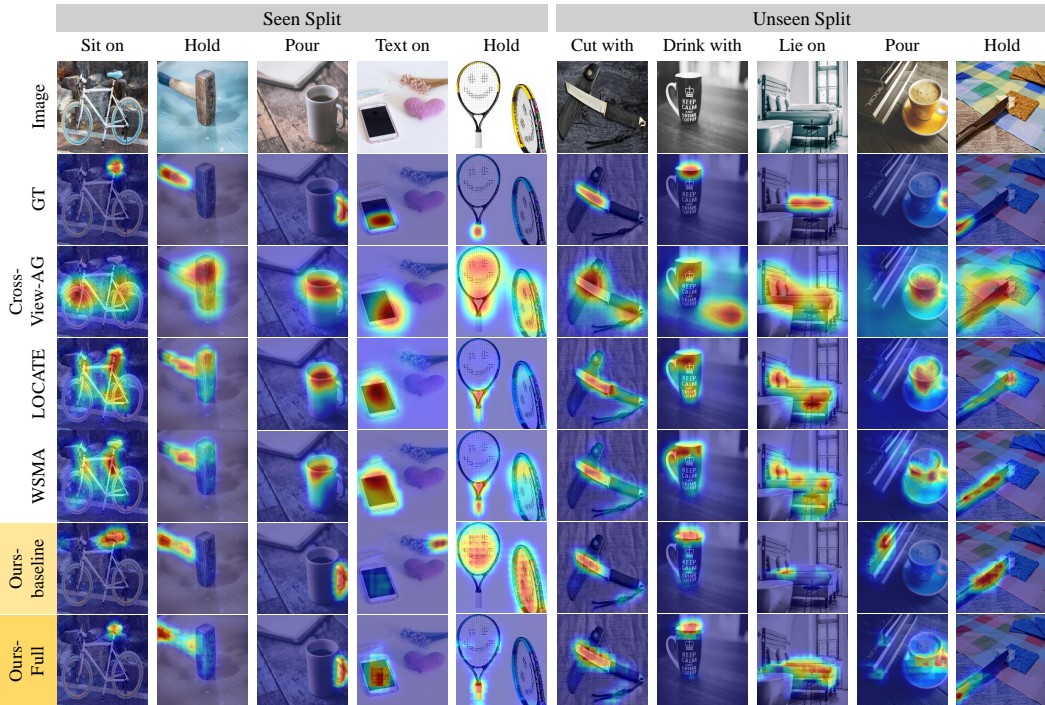

Figure 11: More qualitative comparisons. Thanks to the pseudo label, our baseline model outperforms previous methods on samples like the bicycle for "sit on", the hammer for "hold", the cup for "pour", the knife for "cut with", and the cup for "drink with". Our full model makes further improvements by focusing on the correct object (the cell phone for "text on" and the cup for "pour"), focusing on the correct part (the tennis racket for "hold" and the knife for "hold"), and giving more reliable predictions (the bed for "lie on").

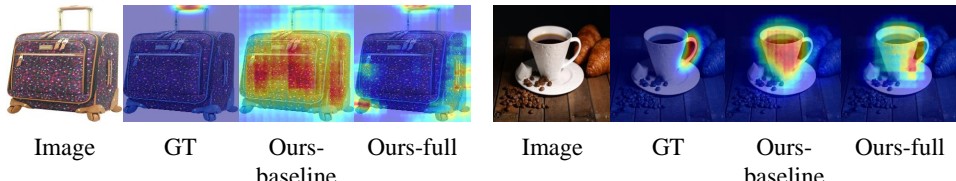

Figure 12: Some typical failure cases of our method. The left example is in the "drag" class of the seen split. Though the full model gives a more concentrated prediction than the baseline model, it still contains clear noise. This is mainly due to that the pseudo labels for some images are still not perfect even after the refinement stage. The right example is in the "pour" class of the unseen split. According to the class distribution in the unseen split, the model has to generalize from bottles and wine glasses to cups, which might be challenging since they usually have very different shapes.

Table 9: Ablation study on the refined pseudo labels. The results are obtained on the seen split where the label refinement stage has a more significant effect.

| | KLD↓ | SIM↑ | NSS↑ |
|---|---|---|---|
| Ours-baseline | 0.938 | 0.503 | 1.477 |
| +refined labels w.o. post-processing | 0.943 | 0.503 | 1.483 |
| +refined labels w. post-processing | **0.916** | **0.506** | **1.524** |

Table 10: Ablation study on the reasoning module. The results are obtained on the unseen split where the reasoning module has a more significant effect.

| part cls | obj cls | stitch aug | KLD↓ | SIM↑ | NSS↑ |
|---|---|---|---|---|---|
| | | | 1.256 | 0.428 | 1.346 |
| ✓ | | | 1.244 | 0.426 | 1.331 |
| ✓ | ✓ | | **1.234** | 0.424 | 1.323 |
| ✓ | ✓ | ✓ | 1.242 | **0.433** | **1.353** |

Table 11: Ablation study on the choice of the visual encoder.

| Encoder | Seen Split | | | Unseen Split | | |
|---|---|---|---|---|---|---|
| | KLD↓ | SIM↑ | NSS↑ | KLD↓ | SIM↑ | NSS↑ |
| CLIP | 0.938 | 0.503 | 1.477 | 1.256 | 0.428 | 1.346 |
| DINO | 0.945 | 0.506 | 1.473 | 1.274 | 0.415 | 1.298 |
| DINOv2 | **0.894** | **0.511** | **1.538** | **1.191** | **0.434** | **1.363** |
| OWL-ViT | 0.957 | 0.491 | 1.451 | 1.239 | 0.416 | 1.334 |
| SAM | 0.999 | 0.482 | 1.421 | 1.304 | 0.394 | 1.253 |

SIM and NSS metrics. After applying the stitching augmentation to stabilize training and mitigate overfitting, the reasoning module achieves consistently improved performance over the base model across all metrics.

Fourthly, we consider different choices of the visual encoder, and the results are shown in Table 11. For each encoder, we use the ViT-B version to make a fair comparison. It can be noted that all the evaluated encoders achieve performance surpassing previous CAM-based methods. In particular, the default choice (CLIP (Radford et al., 2021)) slightly outperforms DINO (Caron et al., 2021) and OWL-ViT (Minderer et al., 2022). This could be attributed to the use of CLIP encoder in the text branch, which enables the affordance query to interact more effectively with the visual features in CLIP's latent space. SAM (Kirillov et al., 2023), on the other hand, exhibits a larger performance gap compared to CLIP. One possible explanation is that SAM's encoder does not incorporate a class token, which hinders the cross-modal fusion from extracting high-level semantic information. Conversely, DINOv2 (Oquab et al., 2024) achieves better results than the default option, likely owing to the favorable properties emerged from self-supervised representation learning. This also aligns with DINOv2's strong performance in dense recognition tasks.

Finally, we try out alternative foundation models for pseudo label generation. Our WSAG pipeline is not tied with VLpart (Sun et al., 2023b) and SAM (Kirillov et al., 2023), and they can be replaced by similar models. In Table 12, VLpart+SAM is our default choice. FastSAM (Zhao et al., 2023) is a YOLO-based lightweight segmentation model which claims 50× higher speed than SAM. Part-GLEE (Li et al., 2025) is a very recent work following the task setting of VLpart. It can be observed that our approach achieves better performance with stronger foundation models. Specifically, using PartGLEE+SAM, the model establishes a new state of the art. Even when using the weaker VL-Part+FastSAM, the model still outperforms previous CAM-based methods. We believe this is an encouraging sign, indicating that the development of general vision models will continue to drive progress in the field of affordance grounding.

Table 12: Ablation study on the choice of the foudation models.

| det. | seg. | Seen Split | | | Unseen Split | | |
|---|---|---|---|---|---|---|---|
| | | KLD↓ | SIM↑ | NSS↑ | KLD↓ | SIM↑ | NSS↑ |
| VLpart | FastSAM | 0.976 | 0.482 | 1.473 | 1.219 | 0.420 | 1.344 |
| VLpart | SAM | 0.890 | 0.510 | 1.547 | 1.153 | 0.437 | 1.418 |
| PartGLEE | SAM | **0.863** | **0.538** | **1.622** | **1.084** | **0.460** | **1.537** |

Table 13: The success rate of grasping with and without the affordance model. Each object is grasped for five attempts, and the object is placed at a random orientation before each grasp.

| | cup | knife | teapot | pan | pressure cook | spoon | wrench | screwdriver | avg |
|---|---|---|---|---|---|---|---|---|---|
| | | | | | Success Rate | | | | |
| w.o. aff | 1/5 | 0/5 | 5/5 | 2/5 | 4/5 | 0/5 | 3/5 | 5/5 | 50.0% |
| w. aff | 3/5 | 3/5 | 4/5 | 4/5 | 4/5 | 4/5 | 4/5 | 5/5 | 77.5% |

Figure 13: Some visualization results for the deployment experiment. In each row, from left to right we show the predicted heatmap for the on-hand camera observation, all the grasps generated by Ma et al. (2024), the selected grasp according to affordance, and an image captured when the gripper grasps the object. Also, we show on the rightmost an image captured when the affordance information is not utilized. It can be seen that the predicted heatmaps help to achieve more stable and more human-like grasps.

### D.4 DETAILS OF THE DEPLOYMENT EXPERIMENT

We aim to validate the real-world performance of our model in the presence of domain gaps, novel objects, and open vocabulary queries through a robot experiment. We deploy the model trained on AGD20K's seen split to an Aubo-i5 robotic arm with a Robotiq 2F-85 two-finger gripper. The observation of the scene is captured by an on-hand Intel RealSense D435i RGB-D camera. We use a local occupancy-enhanced object grasping model (Ma et al., 2024), which takes a single observation of the scene at a fixed camera pose, and generates a set of 6-DoF grasp poses. We then send the RGB observation image to the affordance grounding model, project the grasp points to the image, and select the grasp with the highest value in the affordance heatmap. The selected grasp is then conducted with inverse kinematics.

We conduct robot experiments with 8 different objects. Some representative demonstrations can be found in the supplementary video. The success rate is shown in Table 13. Here, a grasp is treated as a successful one when the gripper manages to lift the object and move it to a given location. Considering affordance, it also has to be located at a proper area on the object, *e.g.* the handle of the knife instead of the blade. Among the 8 categories, only cup and knife are seen during the training process. Our affordance model enhances the grasping quality on both the seen and novel categories. In particular, it is especially helpful for categories such as cup, knife, and spoon, where many positions are geometrically graspable, but only some of them are semantically reasonable. Figure 13 exhibits some examples of the affordance-guided grasping process. Besides, the results in Table 13 are all based on the query "hold", while we also tried substituting it with open-vocabulary synonyms, such as "grasp", and the model outputs similar heatmaps. Real-world experiments involving more diverse affordance queries are left for future research, which may require dexterous hands or equipment with better versatility.

# E  LIMITATIONS AND FUTURE WORKS

Our model currently employs a simple reasoning module based on MLPs. Though it has addressed some generalization issues, its expressive power is still lacking. As in ManipVQA (Huang et al., 2024) and AffordanceLLM (Qian et al., 2024), it might be beneficial to incorporate an LLM into the grounding pipeline to further enhance generalization to novel objects and novel actions. Also, we plan to utilize additional techniques like semantic correlation to improve the pseudo label refinement process, by introducing more local, pixel-level feature information into the masked-pooling-based alignment.

On the other hand, as visual foundation models continue to advance, pixel-level pseudo-labeling for images from the internet or other datasets will become increasingly reliable and convenient. We believe the promising potential in expanding the scale of datasets and increasing the number of object and affordance categories, which would lead to more generalized models. Meanwhile, we observe a certain degree of ambiguity in the definition of affordance within existing datasets, where some affordance categories represent regions where humans can perform actions (*e.g.* hold), while some denote regions where objects can fulfill specific functions (*e.g.* contain). There are already endeavors aimed at addressing fine-grained affordances (Yu et al., 2023; Luo et al., 2023b). In our view, when constructing a larger-scale dataset, it is meaningful to clarify the definition of affordance and perform some form of meta-categorization for different affordance classes. For instance, is the target object used as a tool? Is the affordance related to a task-specific or task-agnostic action (open vs. simply hold)? Is the affordance tied to any human body parts or hand poses?

