# OpenReview forum: "Weakly-Supervised Affordance Grounding Guided by Part-Level Semantic Priors"
_ICLR.cc/2025/Conference — ICLR 2025 Poster_

### Official Review · Reviewer_SXnA · 2024-11-01

**Soundness:** 3
**Presentation:** 3
**Contribution:** 3
**Rating:** 5
**Confidence:** 3

**Summary:**

This paper propose a weakly supervised affordance grounding framework. It uses off-the-shelf foundation models to generate pseudo labels of object parts. To further improve the performance, a label refining strategy, a fine-grained feature alignment process, and a lightweight reasoning module are introduced. Experiments show promising results.

**Strengths:**

1. Training affordance grounding models with object labels is an interesting question.
2. Using off-the-shelf foundation models to generate affordance label is an interesting idea.
3. Experiments show promising results.

**Weaknesses:**

1. As shown in the ablation study table 2, the improvements of using all these three modules look marginal over using one module. It seems that the effectiveness of the three components are not significant.
2. In section 3.4, the authors propose to align the features of exo- and egocentric images after SAM segmentation while the existing methods directly align the features of the two images. However, there is no solid experiments to show the effectiveness of this design.
3. The framework refines the affordance labels with the need of the corresponding exocentric image which may not be available sometimes.

**Questions:**

1. Aligning the features of an object from different views is a commonly used strategy for feature learning. How is this strategy related to pseudo label generation and refinement.
2. Some designs need more detailed ablation studies. E.g., how does the proposed fine-grained feature alignment process with SAM perform when compared with the previous work aligning the features directly. Is there any significant performance difference?

---

> ### Author Response · Authors · 2024-11-23
> **Author Response to Reviewer SXnA**
>
> We are grateful to the reviewer for the valuable insights and suggestions. In the following, we provide point-by-point responses to each concern.
>
> **Marginal improvements of the modules** (Weakness 1)
>
> The performance gain brought by the combination of the three modules is indeed relatively modest. However, we would like to emphasize the complementary nature of the proposed modules. As discussed in Section 4.4, the reasoning module primarily improves the performance on the unseen split (due to enhanced generalization ability), the label refinement stage is particularly beneficial for the seen split (as the refined labels directly contribute to a better understanding of seen object categories), and the alignment loss benefits both splits (due to the robustness brought by introducing exocentric images). From the results in Table 2, each module demonstrates its intended effect, supporting the validity of our design. Moreover, by employing all three modules together, the model can achieve consistent and significant improvements over our baseline model across both splits. Therefore, we believe the design of these three modules is well-justified. As a whole they can make full use of the hints from the exocentric images, the textual input, and the pseudo labels.
>
> **Effects of foundation model aided alignment stage** (Weakness 2 \& Question 2)
>
> We provide in-depth ablation study on the alignment process in Appendix D.3, and the results are shown in Table 8. We copy the table here for ease of reference.
>
> |                          | KLD$\downarrow$ | SIM$\uparrow$ | NSS$\uparrow$ |
> | ------------------------ | --------------- | ------------- | ------------- |
> | Ours-baseline            | 1.256           | 0.428         | 1.346         |
> | +alignment w.o. obj mask | 1.259           | 0.422         | 1.312         |
> | +alignment w. obj mask   | **1.176**       | **0.437**     | **1.407**     |
>
>
> By examining the effect of using the exocentric object masks ($M^{\text{exo}}_{\text{obj}}$) in
> the alignment process, we found that the cross-view feature alignment is not helpful without the masks. Focusing on the object area has a significant advantage over globally pooling the exocentric feature maps, since it excludes the irrelavent information from the alignment process.
>
> In Appendix D.3, we have also conducted ablation studies on the design of the reasoning module, the post-processing strategy of the refinement stage, the choice of the encoder (based on Reviewer 7PPq's suggestion), and the choice of the VFM (based on Reviewer Yqjp's suggestion). Please let us know if there are further concerns.
>
> **Need of exocentric images** (Weakness 3)
>
> The proposed method indeed relies on exocentric images. However, we believe that unlabeled exocentric images, which do not need to be strictly paired with egocentric images (i.e., they may have different object instances), are relatively easy to obtain, especially when compared to pixel-level affordance annotations. For example, such images can be sourced from the internet (by searching for "\[affordance category\] \[object category\]"), video datasets capturing human activities, or generative models.
>
> **Relation with cross-view alignment in feature learning** (Question 1)
>
> To the best of our knowledge, alignment between multiple views is a common strategy in self-supervised learning (e.g., SimCLR[1], DINO[2]). These works leverage the consistency between different augmented views of the same image to learn high-quality visual features. In the context of our task, however, the egocentric/exocentric views (as defined by Cross-View-AG[3]) are unrelated to augmentations. Instead, they refer to two distinct images containing objects of the same category but in different states. Consequently, the view alignment in feature learning is fundamentally different from our alignment process. While the former focuses on consistency under perturbations, the latter aims to transfer specific knowledge from the exocentric view to the egocentric view.
>
> [1] A simple framework for contrastive learning of visual representations. ICML 2020
>
> [2] Emerging Properties in Self-Supervised Vision Transformers. ICCV 2021
>
> [3] Learning affordance grounding from exocentric images. CVPR 2022
>
>
>
> We hope the responses above have addressed the reviewer's concerns, and we are always open to further dicussions.

---

> > ### Comment · Reviewer_SXnA · 2024-11-27
> >
> > Thanks for the authors' response. Some of my concerns are addressed while my concerns on the significance of the combination of the three components  (i.e., weaknesses 1) and question 1 remain. I would like to keep my original score.

---

> > > ### Author Response · Authors · 2024-11-28
> > > **Additional Author Response to Reviewer SXnA**
> > >
> > > We sincerely thank the reviewer for the feedback. Here we provide additional responses to the unresolved concerns.
> > >
> > >
> > > **Significance of combining the three components** (Weakness 1)
> > >
> > > In addition to the intuitive benefits we analyzed in the previous response, we would also quantitatively emphasis the significance of combining all three modules. Below is a brief summary for the information provided in Table 2 of the main text, where the best-performing single module (i.e., the alignment module) is compared with the combination. The results are averaged across the seen and unseen splits.
> > >
> > > |                    | KLD$\downarrow$   | SIM$\uparrow$     | NSS$\uparrow$     |
> > > | ------------------ | ----------------- | ----------------- | ----------------- |
> > > | baseline           | 1.097             | 0.466             | 1.412             |
> > > | +alignment module  | 1.050 (-4.3\%)     | 0.472 (+1.4\%)     | 1.446 (+2.5\%)     |
> > > | +all three modules | **1.022 (-6.9\%)** | **0.474 (+1.7\%)** | **1.482 (+5.0\%)** |
> > >
> > > It can be observed that the full model achieves a 6.9\% improvement on KLD and a 5.0\% improvement on NSS over the baseline. Nonetheless, using any individual module leads to at most 4.3\% and 2.5\% improvement, respectively. We argue that these performance gaps imply non-trivial gains.
> > >
> > > **About cross-view alignment** (Question 1)
> > >
> > > We fully agree with the reviewer that cross-view learning is common, but strongly argue that it does not serve as a basis to reject this work. Our previous response has elaborated on the differences between view alignment in the literature of self-supervised learning and our interested task. Here we would further clarify that the alignment of egocentric / exocentric images is actually a default setting in weakly supervised affordance grounding (Lines 84-91, Lines 266-275). Though the alignment per se does not imply any technical contribution, designing the scheme of extracting most informative bits from exocentric images is non-trivial. We believe that the proposed foundation model-based scheme (Section 3.4), as validated in Appendix D.3 (Table 8), makes solid and novel addition to this task, and we hope the reviewer could re-evaluate this work with this clarification.
> > >
> > >
> > > **The relationship between cross-view alignment and pseudo labels** (Question 1)
> > >
> > > "Aligning egocentric and exocentric images" and "utilizing pseudo labels" are two orthogonal ideas in affordance grounding. This work explored both ideas in the proposed model, and each of them proves to be clearly beneficial.
> > >
> > > We appreciate the time and effort devoted to reviewing this paper, and we always welcome further discussions.

---

### Official Review · Reviewer_Yqjp · 2024-11-04

**Soundness:** 3
**Presentation:** 3
**Contribution:** 3
**Rating:** 6
**Confidence:** 3

**Summary:**

This paper addresses the task of weakly supervised affordance grounding (WSAG), where the goal is to identify affordance regions on objects using only image-level labels and human-object interaction images.
The key contributions include:
- A novel pseudo-supervised training framework and pipeline that leverages visual foundation models to generate affordance heatmaps, mapping affordance classes to object parts.
- Three key enhancements to improve performance:
    - Label refinement using interaction cues
    - Fine-grained object feature alignment with exocentric images
    - Reasoning module for better generalization
- Extensive experiments demonstrating significant performance improvements over existing methods

**Strengths:**

- Clear writing and organization.
- Well-motivated technical approach with clear problem formulation.
- This paper propose a novel approach that uses visual foundation models and part-level semantic priors for WSAG, unleashing the power of these models for affordance learning.
- Using human occlusion cues for label refinement, which is an innovative insight.
- Comprehensive experimental validation and thoughtful analysis of limitations in existing methods.

**Weaknesses:**

- Could benefit from more analysis of failure cases.
- The label refinement stage using human occlusion cues may be problematic when interactions are ambiguous or when multiple affordances exist.
- The mapping from affordance to part names is ad-hoc and manually crafted, which limits the scalability to new affordance types and more complex objects.

**Questions:**

1. Could you provide more details about failure cases and limitations of the proposed approach?
2. How sensitive is the method to the results of VFM? How well can the refine state correct possible errors by VLpart and SAM?
3.  How does the computational cost (training & inference) compare to existing CAM-based methods?

---

> ### Author Response · Authors · 2024-11-23
> **Author Response to Reviewer Yqjp (Part 1)**
>
> We sincerely appreciate the reviewer's constructive and thorough feedback. Below, we provide detailed responses to the reviewer's comments.
>
> **Failure cases analysis and limitations** (Weakness 1 \& Question 1)
>
> We present some examples of the failure cases in Figure 12 in the Appendix. As detailed in the caption of Figure 12, we observe two kinds of typical failure cases. First, for some small or intricate affordance regions (e.g. the part of a suitcase that affords dragging), the pseudo labels, even after refinement, are not accurate enough. Second, for some objects in the unseen test set that exhibit significant differences in shape or structure compared to the training objects (e.g., holding a cup vs. holding a wine glass), the model's generalization ability still requires improvement.
>
> We kindly refer the reviewer to Appendix E for our discussion on limitations and future directions. In brief, possible improvements include: incorporating external knowledge to enhance reasoning capabilities on novel categories, introducing finer-grained image correlations to improve the alignment module, scaling up the dataset size, and establishing a more principled taxonomy for affordance.
>
> **Problem of the label refinement stage** (Weakness 2)
>
> The reviewer is correct that the label refinement process may meet difficulty when faced with complex affordances and interactions. In the refinement stage, we aim to focus on the affordance categories that are related to human body, especially hands. Exocentric images of these categories (e.g. hold, hit, drag) usually contain clear human-object occulusions that are informative for our model. It is also worth noting that hand-related affordances hold particularly significant practical value for current research in embodied intelligence, as they are directly related to object manipulation.
>
> **Ad-hoc part name mapping** (Weakness 3)
>
> As stated in Line 240-241, the manually constructed part name mapping is sufficient for the experiments on AGD20K, while implementing an automatically generated mapping is also feasible. In Table 5 (in Appendix), we present some examples of such a mapping generated by GPT-4o and the prompt we use. The results align well with the handcrafted mapping.

---

> ### Author Response · Authors · 2024-11-23
> **Author Response to Reviewer Yqjp (Part 2)**
>
> **Sensitivity to VFMs and effect of the refinement stage** (Question 2)
>
> We would like to address the sensitivity to VFMs from the following perspectives.
>
> (1) It is important to choose the right type of VFMs in the first place.
>
> In Appendix B.1, we explore three different ways of using VFMs for pseudo label generation, including directly using a multimodal LLM, directly using a foundation model specialized for detection, and combining a detection model with label mapping. From the results in Figure 6 and 7, we observe that only the third approach can produce reasonable labels in most cases. Therefore, the first step in using VFMs is to design a proper pipeline to translate its semantic priors into affordance-related information.
>
> (2) The errors in the initial pseudo labels can be fixed.
>
> Given the relatively high-quality pseudo labels, our approach can correct the errors within them through two mechanisms, as shown in Figure 9 and 10 in the Appendix. On one hand, the supervised training pipeline will fix some **occasional errors**. Though the pseudo labels may miss some instances or deviate from the correct region, the trained model can make correct predictions because a lot of samples in the same category have correct labels (Figure 9). On the other hand, the label refinement stage will deal with some **systematic errors**, i.e. the cases where the VFM produces incorrect labels for the majority of images within certain category. In Figure 10 we visualize the progress made by the refinement stage. These results effectively demonstrate the robustness of our approach to the errors of VFMs.
>
>
> (3) The overall performance will increase with the capability of the VFMs.
>
> Our pipeline is not tied with VLpart and SAM, and they can be replaced by similar models. To validate this, we perform an additional ablation study on the choice of VFMs for label generation.
>
> |          |         |                 | Seen          |               |                 | Unseen        |               |
> | -------- | ------- | --------------- | ------------- | ------------- | --------------- | ------------- | ------------- |
> | det.      | seg.     | KLD$\downarrow$ | SIM$\uparrow$ | NSS$\uparrow$ | KLD$\downarrow$ | SIM$\uparrow$ | NSS$\uparrow$ |
> | VLpart   | FastSAM | 0.976           | 0.482         | 1.473         | 1.219           | 0.420         | 1.344         |
> | VLpart   | SAM     | 0.890           | 0.510         | 1.547         | 1.153           | 0.437         | 1.418         |
> | PartGLEE | SAM     | **0.863**       | **0.538**     | **1.622**     | **1.084**       | **0.460**     | **1.537**     |
>
> Here, VLpart+SAM is our default choice. FastSAM [1] is a YOLO-based lightweight segmentation model which claims 50× higher speed than SAM. PartGLEE [2] is a very recent work following the task setting of VLpart. It can be observed that our approach achieves better performance with stronger foundation models. Specifically, using PartGLEE+SAM, the model establishes a new state of the art. Even when using the weaker VLPart+FastSAM, the model still outperforms previous CAM-based methods. We believe this is an encouraging sign, indicating that the development of general vision models will continue to drive progress in the field of affordance grounding.
>
> This experiment will be added to Appendix D.3 in our revised version.
>
> [1] Fast Segment Anything. https://arxiv.org/pdf/2306.12156
>
> [2] PartGLEE: A Foundation Model for Recognizing and Parsing Any Objects. ECCV 2024
>
> **Computational cost analysis** (Question 3)
>
> We provide an analysis on the efficiency of our model in Appendix A.4, and the model statistics are listed in Table 3 and 4. In brief, our model has comparable inference speed (\~49 fps) with previous methods like Cross-View-AG (\~52 fps) and WSMA (\~30 fps), and the computation is mainly concentrated at the visual encoder.
>
> As for training, it takes about 1h to generate the initial pseudo labels for the training set. The label refinement stage requires approximately 1h, followed by around 3h for supervised training. The whole training process can be performed on a single NVIDIA GeForce RTX 2080Ti. As reference, LOCATE's training scheme takes about 7h in the same environment, while WSMA takes about 2h. So the training cost of our method is basically at the same level with previous methods. Besides, our supervised training process is more straightforward than previous works, involving neither the clustering operations used by LOCATE nor the non-negative matrix factorization employed in WSMA. (This paragraph will be added to Appendix A.4 in our revised version.)
>
>
>
> We hope the responses above have addressed the reviewer's concerns, and we are always open to further dicussions.

---

> > ### Comment · Reviewer_Yqjp · 2024-11-27
> >
> > Thank you for the detailed response, I will keep my score.

---

### Official Review · Reviewer_7PPq · 2024-11-04

**Soundness:** 4
**Presentation:** 4
**Contribution:** 3
**Rating:** 8
**Confidence:** 4

**Summary:**

This paper tackles weakly supervised affordance grounding (WSAG) by leveraging foundation models to generate pseudo labels, departing from previous CAM-based approaches. The authors propose a three-stage pipeline: (1) using VLpart and SAM to generate initial pseudo labels by mapping affordance-object pairs to part names, (2) refining these labels using human-object interaction cues from exocentric images, and (3) training an affordance grounding model with the refined pseudo labels. The method also includes cross-view feature alignment and a reasoning module to handle unseen objects. The approach shows significant improvements over existing WSAG methods

**Strengths:**

- The problem is important and well-motivated, as affordance grounding is crucial for robotic manipulation and human-object interaction understanding
- The proposed pseudo-labeling approach effectively leverages existing foundation models (VLpart, SAM) to provide supervision, addressing limitations of previous CAM-based methods
- The label refinement process using exocentric images is novel and well-designed, providing a clever way to improve initial pseudo labels
- The reasoning module helps generalize to unseen objects, which is crucial for practical applications
- The writing is clear and the method is well-explained with appropriate visualizations

**Weaknesses:**

The choice of CLIP as the vision encoder could be better justified given previous work suggesting limitations (vs DINO, OWLViT, SAM). For example, the paper will be stronger with an ablation study of different visual encoders.

**Questions:**

See weaknesses.

---

> ### Author Response · Authors · 2024-11-23
> **Author Response to Reviewer 7PPq**
>
> We sincerely thank the reviewer for the time and effort, and we are very grateful for the constructive and positive review. In response to the reviewer's suggestion, we have performed an additional ablation study on the visual encoder.
>
> |         |                 | Seen          |               |                 | Unseen        |               |
> | ------- | --------------- | ------------- | ------------- | --------------- | ------------- | ------------- |
> | Encoder | KLD$\downarrow$ | SIM$\uparrow$ | NSS$\uparrow$ | KLD$\downarrow$ | SIM$\uparrow$ | NSS$\uparrow$ |
> | CLIP    | 0.938    | 0.503         | 1.477  | 1.256           | 0.428  | 1.346  |
> | DINO    | 0.945           | 0.506  | 1.473         | 1.274           | 0.415         | 1.298         |
> | DINOv2  | **0.894**       | **0.511**     | **1.538**     | **1.191**       | **0.434**     | **1.363**     |
> | OWL-ViT | 0.957           | 0.491         | 1.451         | 1.239    | 0.416         | 1.334         |
> | SAM     | 0.999           | 0.482         | 1.421         | 1.304           | 0.394         | 1.253         |
>
> For each encoder, we use the ViT-B version to make a fair comparison. It can be observed that all the evaluated encoders achieve performance surpassing previous CAM-based methods. Specifically, the default choice (CLIP) slightly outperforms DINO and OWL-ViT. This could be attributed to the use of CLIP encoder in the text branch, which enables the affordance query to interact more effectively with the visual features in CLIP's latent space. SAM, on the other hand, exhibits a larger performance gap compared to CLIP. One possible explanation is that SAM's encoder does not incorporate a class token, which hinders the cross-modal fusion (detailed in Section 3.2 and Appendix A.1) from extracting high-level semantic information. Conversely, DINOv2 achieves better results than the default option, likely owing to the favorable properties emerged from self-supervised representation learning. This also aligns with DINOv2's strong performance in dense recognition tasks.
>
> In summary, our pipeline is well compatible with different visual encoders. As our goal is to establish a general framework that leverages foundation models for the affordance task, selecting the optimal visual encoder falls beyond the scope of this paper. However, we do observe that more advanced encoders yield superior overall performance, and we appreciate the reviewer's insightful remarks in this regard.

---

> > ### Comment · Reviewer_7PPq · 2024-11-26
> >
> > Thanks for the detailed experiments! That totally makes sense to me.

---

### Official Review · Reviewer_dPdq · 2024-11-19

**Soundness:** 3
**Presentation:** 3
**Contribution:** 3
**Rating:** 6
**Confidence:** 3

**Summary:**

The proposed framework for weakly supervised affordance grounding (WSAG) uses pseudo-supervised learning to link affordance actions to object parts via part segmentation models and semantic cues. It generates and refines pseudo-labels by focusing on affordance-relevant regions with exocentric images, improving label accuracy and feature alignment. To enhance generalization, a lightweight reasoning module maps affordances to latent object part representations, enabling the model to handle unseen categories. By integrating semantic knowledge from foundation models, the framework transitions from weakly to pseudo-supervised learning, achieving a breakthrough in performance over prior methods

**Strengths:**

1) The paper is clearly written and easy to follow.
2) The method is well-motivated, and the VFM-assisted pseudo-labeling should effectively address the challenges of the weakly-supervised setting.
3) The overall improvements over existing methods are quite significant.

**Weaknesses:**

My biggest concern lies in the experimental section. In Table 2, the reasoning model appears to negatively impact the baseline, and the other two design components only provide marginal improvements.

**Questions:**

Could the authors clarify why the baseline method in Table 2 outperforms existing state-of-the-art methods by such a significant margin? Based on the numbers in Tables 1 and 2, it seems the improvement over existing methods might primarily stem from a strong baseline, while the additional modules contribute only marginal benefits

---

> ### Author Response · Authors · 2024-11-23
> **Author Response to Reviewer dPdq**
>
> We sincerely appreciate the reviewer for providing a prompt and thorough emergency review. Here is our response to the reviewer's concerns.
>
> **Clarification of the baseline**
>
> The "baseline" method in Table 1 refers to supervised training with the initial pseudo labels generated by the foundation models, i.e., the method described in Section 3.3. We would like to make it clear that the strong baseline itself is part of our contributions. Its improvement over previous methods is primarily attributed to the shift from CAM-based prediction to supervised training using pseudo labels. As mentioned in Line 77-84, the CAM is focused on the most discriminative part of the image, while it may fail to accurately capture the affordance region associated with an action. In contrast, the pseudo labels generated by the foundation models are of higher quality, and enable a consistent pipeline for training and inference (as shown in Figure 1(c)).
>
> **Improvements of the proposed modules**
>
> In addition to the baseline method, the other part of our contribution lies in the design of the three extra modules. We would like to emphasize that these modules have necessary and complementary effects, as detailed in Section 4.4. The reasoning module is designed for improving generalization. Thus, as observed by the reviewer, it does not show advantage on the seen split (better SIM but worse KLD and NSS), where the model has encountered all kinds of objects during training and the test set does not demand cross-category generalization. On the unseen split, where generalization ability is essential, including the reasoning loss consistently enhances the performance on all metrics. Similarly, the refinement stage has more significant effect for the seen split, since the refined labels directly contribute to a better understanding of seen categories. The cross-view alignment works well for both splits, showing the benefits of introducing exocentric images to guide feature learning.
>
> Also, from a more quantitative perspective, the improvements achieved by incorporating the three modules into the baseline model (e.g., -0.05 KLD on the Seen split and -0.10 KLD on the Unseen split) are comparable to the improvements brought by prior methods (e.g., Cross-View-AG+ vs. Cross-View-AG, WSMA vs. LOCATE). Overall, we believe that the design of these modules is necessary and meaningful.
>
> We hope the content above addresses the reviewer's concerns regarding our experimental results, and we are always open to further dicussions.

---

### Author Response · Authors · 2024-11-23
**Author Response to All Reviewers**

We would like to thank all the reviewers for the efforts they devoted and the suggestions they gave. In the rebuttal, we have provided responses and clarifications regarding the effect of the proposed modules, detailed ablation studies, computational cost, failure cases analysis, task setup, and connections to related topics. Meanwhile, we have made the following modifications to the paper (highlighted in red):
- A new paragraph in Appendix A.4 describing the computational cost of training.
- Two new experiments in Appendix D.3 on the choice of visual encoders and foundation models. (With the help of more advanced encoder and foundation models, we obtain even stronger performance than the originally reported ones!)

We hope that these contents could address the reviewers' concerns and present our work more clearly. If we have not fully addressed your questions or if you have any further inquiries, please do not hesitate to contact us.

---

### Meta-Review · Area_Chair_eaNR · 2024-12-20

**Metareview:**

This paper introduces a VFM-assisted pseudo-labeling method to address the weakly-supervised affordance grounding task. The proposed approach incorporates three key modules: (1) a label refining strategy using exocentric images, (2) a fine-grained feature alignment process with exocentric images, and (3) a lightweight reasoning module. These modules collectively improve performance on the task. The use of exocentric images in both label refining and feature alignment is particularly novel and provides valuable insights. The primary concern is the marginal improvement in performance relative to the baseline method.

Initial reviewer concerns focused on several aspects of the proposed method, including:

- The superior performance of the baseline method (dPdq) and the marginal improvements from the proposed modules (dPdq, SXnA)
- The ablation study on different visual encoders (7PPq)
- Failure cases and limitations (Yqjp)
- Sensitivity to VFM results (Yqjp)
- The ad-hoc nature of the affordance-to-part names mapping (Yqjp)
- Computational complexity (Yqjp)
- The necessity of exocentric images (SXnA)
- The novelty of cross-view alignment and its relationship to pseudo-label generation (SXnA)

The authors have actively engaged with each of these concerns in their rebuttal. Most reviewers acknowledged that their issues had been addressed, leading to a positive shift in ratings: 8, 6, 6, and 5. However, Reviewer SXnA still expressed reservations regarding the marginal improvements of the proposed modules and the relationship between cross-view alignment and pseudo-label refinement.

After carefully reviewing the paper, the reviewers' comments, and the authors' responses, the AC agrees with reviewers dPdq, 7PPq, and Yqjp that the VFM-assisted pseudo-labeling method is both effective and novel. The integration of the three proposed modules on top of the pseudo-labeling baseline leads to consistent performance improvements. Regarding Reviewer SXnA’s concern, the AC agrees with the authors’ explanation that cross-view alignment and pseudo-label refinement are distinct processes and not directly related.

Given the positive ratings from most reviewers and the resolution of concerns - particularly regarding the novelty and effectiveness of the proposed method - the AC recommends accepting this paper for publication.

**Additional Comments On Reviewer Discussion:**

Initial reviewer concerns focused on several aspects of the proposed method, including:

- The superior performance of the baseline method (dPdq) and the marginal improvements from the proposed modules (dPdq, SXnA)
- The ablation study on different visual encoders (7PPq)
- Failure cases and limitations (Yqjp)
- Sensitivity to VFM results (Yqjp)
- The ad-hoc nature of the affordance-to-part names mapping (Yqjp)
- Computational complexity (Yqjp)
- The necessity of exocentric images (SXnA)
- The novelty of cross-view alignment and its relationship to pseudo-label generation (SXnA)

The authors have actively engaged with each of these concerns in their rebuttal. Most reviewers acknowledged that their issues had been addressed, leading to a positive shift in ratings: 8, 6, 6, and 5. However, Reviewer SXnA still expressed reservations regarding the marginal improvements of the proposed modules and the relationship between cross-view alignment and pseudo-label refinement.

After carefully reviewing the paper, the reviewers' comments, and the authors' responses, the AC agrees with reviewers dPdq, 7PPq, and Yqjp that the VFM-assisted pseudo-labeling method is both effective and novel. The integration of the three proposed modules on top of the pseudo-labeling baseline leads to consistent performance improvements. Regarding Reviewer SXnA’s concern, the AC agrees with the authors’ explanation that cross-view alignment and pseudo-label refinement are distinct processes and not directly related.

---

### Decision · Program_Chairs · 2025-01-22

Accept (Poster)